# Suppression of ferroptosis by vitamin A or radical-trapping antioxidants is essential for neuronal development

Juliane Tschuck [1,11], Vidya Padmanabhan Nair[2,11], Ana Galhoz[3,4], Carole Zaratiegui[5], Hin-Man Tai[2], Gabriele Ciceri [6], Ina Rothenaigner[1], Jason Tchieu [6,10], Brent R. Stockwell [7], Lorenz Studer [6], Daphne S. Cabianca[5], Michael P. Menden [3,8], Michelle Vincendeau [2,9,12] ✉ & Kamyar Hadian [1,12] ✉

The development of functional neurons is a complex orchestration of multiple signaling pathways controlling cell proliferation and differentiation. Because the balance of antioxidants is important for neuronal survival and development, we hypothesized that ferroptosis must be suppressed to gain neurons. We find that removal of antioxidants diminishes neuronal development and laminar organization of cortical organoids, which is fully restored when ferroptosis is inhibited by ferrostatin-1 or when neuronal differentiation occurs in the presence of vitamin A. Furthermore, iron-overload-induced developmental growth defects in *C. elegans* are ameliorated by vitamin E and A. We determine that all-trans retinoic acid activates the Retinoic Acid Receptor, which orchestrates the expression of anti-ferroptotic genes. In contrast, retinal and retinol show radical-trapping antioxidant activity. Together, our study reveals an unexpected function of vitamin A in coordinating the expression of essential cellular gatekeepers of ferroptosis, and demonstrates that suppression of ferroptosis by radical-trapping antioxidants or by vitamin A is required to obtain mature neurons and proper laminar organization in cortical organoids.

During brain development, neurons arise in a tightly controlled process regulating an important balance between proliferation and differentiation events[1]. To ensure that cells emerge, migrate, and mature at the right time and space, a network of genetic and molecular factors is required[2]. However, the exact factors and their precise interplay are still not fully understood. Regulated cell death (RCD) eliminates half of the neurons initially generated during mammalian brain development and helps shape the organization of cortical circuits by regulating their

[1]Research Unit Signaling and Translation, Helmholtz Zentrum München, Neuherberg, Germany. [2]Endogenous Retrovirus Group, Institute of Virology, Helmholtz Zentrum München, Neuherberg, Germany. [3]Computational Health Center, Helmholtz Zentrum München, Neuherberg, Germany. [4]Department of Biology, Ludwig-Maximilians University Munich, Munich, Germany. [5]Institute of Functional Epigenetics, Helmholtz Zentrum München, Neuherberg, Germany. [6]Developmental Biology and Center for Stem Cell Biology, Memorial Sloan Kettering Cancer Center, New York, NY, USA. [7]Department of Biological Sciences, Department of Chemistry, Herbert Irving Comprehensive Cancer Center, Irving Institute for Cancer Dynamics, Columbia University, New York, NY, USA. [8]Department of Biochemistry and Pharmacology, University of Melbourne, Parkville Victoria, Australia. [9]Technical University of Munich, Institute of Virology, School of Medicine, Munich, Germany. [10]Present address: UC Department of Pediatrics, Division of Developmental Biology, Cincinnati Children's Hospital Medical, Cincinnati, OH, USA. [11]These authors contributed equally: Juliane Tschuck, Vidya Padmanabhan Nair. [12]These authors jointly supervised this work: Michelle Vincendeau, Kamyar Hadian. ✉e-mail: michelle.vincendeau@helmholtz-munich.de; kamyar.hadian@helmholtz-munich.de

cellular composition[3,4]. Apoptosis, a well-studied RCD modality, has so far been described as the major form of cell death during the developing nervous system[3,4].

Ferroptosis is another RCD, which occurs through iron-dependent lipid peroxidation[5–8]. It is controlled by a number of cellular gatekeepers, including the system-xc-/glutathione peroxidase 4 (GPX4)/glutathione axis[7], the ferroptosis suppressor protein 1 (FSP1)/ubiquinol/vitamin K axis[9–11], the GTP cyclohydrolase 1 (GCH1)/tetrahydrobiopterin (BH4)/dihydrofolate reductase (DHFR) axis[12,13], and the nuclear receptor Farnesoid X receptor (FXR)[14,15], amongst others. Ferroptosis can be the cause of several degenerative diseases of the adult brain, heart, lung, and kidney[7]. However, our understanding of the functions of ferroptosis during brain development remains rudimentary. In this study, we illuminate the detrimental role of increased ferroptosis during neurogenesis and the necessity to inhibit ferroptosis by radical-trapping antioxidants or vitamin A to obtain cortical neurons.

## Results

### Vitamin A restores neuronal differentiation in the absence of antioxidant protection

Most neuronal differentiation protocols require a series of antioxidants (AO) (Supplementary Data 1), including vitamin E and glutathione (GSH), two well-known ferroptosis inhibitors[6], for proper neuronal development. Thus, to understand whether the exclusion of these antioxidants would affect neurogenesis due to extensive ferroptosis and whether vitamin A could compensate for the antioxidant deficiency, we differentiated human embryonic stem cells (H9; WA09) into cortical neurons using an established differentiation protocol[16–18]. It is reported that the concentration of all-trans retinoic acid (ATRA) (active vitamin A metabolite) is important for its distinct functions[19,20]. Hence, we used three conditions: without or with increasing vitamin A concentrations between day 10 to day 20 of differentiation (Fig. 1a): (i)

standard media with antioxidant-containing B27 without vitamin A (+AO; Supplementary Data 1), (ii) media with antioxidant-deficient B27 containing vitamin A (-AO+vA^lo; Supplementary Data 1), or (iii) media with antioxidant-deficient B27 containing vitamin A, which was further supplemented with ATRA (-AO+vA^hi). We performed total RNA sequencing of the +AO and -AO+vA^lo samples and showed that vitamin A supplementation led to overexpression of Retinoic Acid Receptor B (RARB) and beta-carotene oxygenase 2 (BCO2) at day 20 (immature neurons) in -AO+vA^lo treated cells compared to +AO treated cells (Supplementary Fig. 1a, b). RARB is a nuclear receptor employing vitamin A (ATRA) as a ligand for transcriptional activity[21], and BCO2 is an enzyme involved in retinoic acid biosynthesis[22]. Increased expression of RARB and BCO2 at day 20 in -AO+vA^lo treated cells could be validated using qRT-PCR (Supplementary Fig. 1c). We also checked levels of RARB upon low and high exposure to vitamin A and saw a dose-dependent RARB expression (Fig. 1b). Interestingly, lack of antioxidants had the consequence that immature neurons at day 20 of differentiation showed higher lipid peroxidation as detected by the C11-BODIPY lipid peroxidation sensor. However, higher levels of vitamin A could abrogate lipid peroxidation (Fig. 1c). This observation was verified through staining against 4-HNE (a product of lipid peroxidation), where lack of antioxidants generated higher levels of 4-HNE, and vitamin A could reduce the 4-HNE levels (Supplementary Fig. 2a). We then differentiated the three conditions into day 40 cortical neurons and demonstrate that -AO+vA^lo treated conditions showed fewer number MAP2-positive neurons (Fig. 1d). MAP2a/b isoforms are neuron-specific cytoskeletal proteins and a suitable marker for neuronal cells to investigate morphological (cytoskeletal) changes[18]. However, higher levels of vitamin A (-AO+vA^hi) restored the number of MAP2-positive cortical neurons (Fig. 1d). Interestingly, Ki67 staining as a marker of proliferation showed that conditions of low antioxidants had no effect on cell proliferation (Supplementary Fig. 2b). Consistent with this, the neurons that were generated under

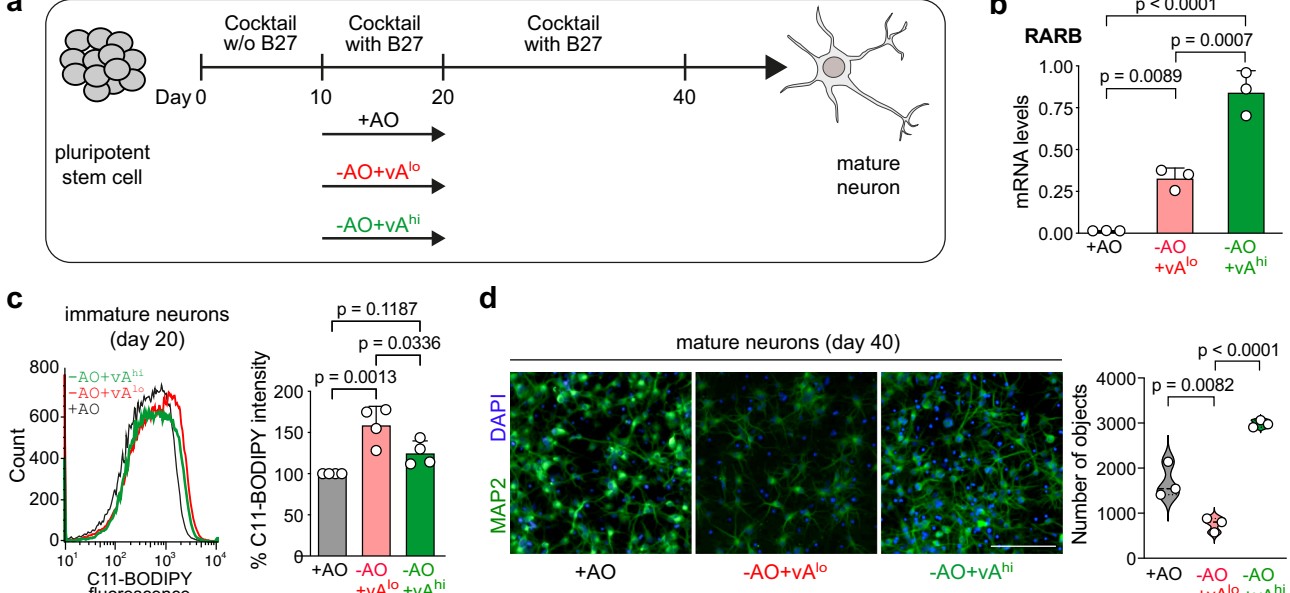

**Fig. 1 | Vitamin A restores neuronal differentiation by reducing lipid peroxidation. a** Scheme of neuronal differentiation using three conditions (i) standard media with antioxidant-containing B27 (+AO), (ii) media with antioxidant-deficient B27 containing vitamin A (-AO+vA^lo), or (iii) media with antioxidant-deficient B27 containing vitamin A, which was further supplemented with retinoic acid (RA) (-AO+vA^hi). **b** mRNA levels of RARB at day 20 of cortical neuronal differentiation using quantitative RT-PCR. Data are mean ± SD of $n = 3$ biologically independent replicates, one-way ANOVA with Tukey's test. **c** C11-BODIPY staining of day 20

immature cortical neurons supplemented with vitamin A using flow cytometry ($n = 4$). (Left) flow cytometry histograms; (right) median intensity. Data are mean ± SD of $n = 4$ biologically independent replicates; one-way ANOVA with Tukey's test. **d** (Left) MAP2 immunofluorescence staining of cortical neurons at day 40. Scale bars, 100 μm. (Right) high-content image analyses. $n = 3$ biologically independent replicates with every 2 technical replicates; one-way-ANOVA with Tukey's test. Source data are provided as a Source Data file.

-AO + vA$^{lo}$ conditions showed similar levels of mean neurite length and branching (Supplementary Fig. 2c), suggesting that those cells that were able to generate mature neurons under low antioxidant conditions, possibly due to intrinsic upregulation of ferroptosis gatekeepers, have a normal phenotype. This underscores that ferroptotic cell death occurs under antioxidant-deficient conditions, rather than impaired neuronal differentiation or delayed cell proliferation.

## Specific inhibition of ferroptosis promotes neuronal differentiation

To verify whether cells in the absence of antioxidants indeed undergo ferroptosis during neuronal differentiation, we used an established ferroptosis inhibitor, i.e., ferrostatin-1 (Fer-1)[5,7], to supplement for lack of antioxidants to inhibit ferroptosis. Hence, we differentiated H9 cells into cortical neurons using three differentiation conditions (Fig. 2a). Between day 10–20 of differentiation, we used (i) the standard B27-supplemented medium containing antioxidants but not vitamin A (+ AO), (ii) medium with antioxidant-deficient B27 containing low amounts of vitamin A (-AO + vA$^{lo}$), or (iii) medium with antioxidant-deficient B27 containing low amounts of vitamin A, which was further supplemented with ferrostatin-1 (-AO + vA$^{lo}$ + Fer-1) (Fig. 2a). At day 20 of cortical neuronal differentiation, we investigated the level of lipid peroxidation using the C11-BODIPY sensor (Fig. 2b). Cells that differentiated without antioxidants (-AO + vA$^{lo}$) showed an increase of lipid peroxides compared to immature neurons differentiated under standard conditions (+ AO) (Fig. 2b). Similarly, cells differentiated without antioxidants (-AO + vA$^{lo}$) showed higher levels of 4-HNE. Importantly, treatment of the cells with ferrostatin-1 (-AO + vA$^{lo}$ + Fer-1) reverted lipid peroxidation to control levels (Fig. 2b and Supplementary Fig. 2a). Moreover, we differentiated the three conditions into mature cortical neurons. Day-40 cortical neurons were analyzed for neuronal morphology using MAP2 immunofluorescence staining (Fig. 2c). Neurons differentiated without antioxidants and low levels of vitamin A (-AO + vA$^{lo}$) showed a decrease in MAP2 expression as well as fewer neuronal segments (regions of the neuronal outreach) when compared to neurons differentiated in the presence of antioxidants (+ AO) (Fig. 2c). Addition of Fer-1 to the differentiation without antioxidants (-AO + vA$^{lo}$ + Fer-1) fully rescued the observed phenotype (Fig. 2c). Importantly, the number of healthy cells was reduced without antioxidants (determined by healthy nuclei). As the ferroptosis-specific inhibitor Fer-1 fully restored MAP2-positive cells, these data suggest the occurrence of ferroptotic cell death during early events of differentiation in the absence of antioxidants rather than proliferative effects. An effect on proliferation could also be excluded as Ki67 levels did not change in the absence of antioxidants compared to the control (Supplementary Fig. 2b). Analysis of *MAP2* mRNA expression of day-40 cortical neurons confirmed reduced *MAP2* mRNA levels in -AO + vA$^{lo}$ treated cells and a rescue of *MAP2* mRNA levels upon Fer-1 treatment (-AO + vA$^{lo}$ + Fer-1) (Fig. 2d).

We further investigated the role of ferroptosis on neuronal development in a more physiological context by using forebrain organoids. These organoids were generated using a medium containing + AO, -AO + vA$^{lo}$, or -AO + vA$^{lo}$ + Fer-1. Interestingly, organoids without antioxidants (-AO + vA$^{lo}$) exhibited a significantly reduced size at day 40 (as demonstrated by the analysis of the area of the organoids) compared to + AO organoids, or organoids rescued with ferrostatin-1 (-AO + vA$^{lo}$ + Fer-1) (Fig. 2e). We recently demonstrated that the transferrin receptor (TfR1) can be utilized to visualize ferroptotic events[23,24] and as such serves as a ferroptosis marker. Therefore, we sectioned organoids generated in + AO, -AO + vA$^{lo}$ and -AO + vA$^{lo}$ + Fer-1 conditions at day 60 and stained these with antibodies against TfR1. Positive TfR1 staining could only be observed and quantified in -AO + vA$^{lo}$ treated organoid sections, but was undetectable in + AO and -AO + vA$^{lo}$ + Fer-1 conditions, respectively (Fig. 2f), indicating that ferroptosis is present in conditions lacking antioxidants

and can be eliminated when these organoids are treated with the ferroptosis-specific inhibitor Fer-1. In addition, the organoids (+ AO, -AO + vA$^{lo}$ and -AO + vA$^{lo}$ + Fer-1) were sectioned and stained with an anti-4-HNE antibody to detect lipid peroxidation. Similar to the anti-TfR1 staining, organoids without antioxidants showed increased levels of 4-HNE, which was eliminated upon Fer-1 treatment (Fig. 2g). Finally, we took three organoids per condition (+ AO, -AO + vA$^{lo}$ and -AO + vA$^{lo}$ + Fer-1), then dissociated these organoids into single cells and pooled cells from each condition. These cells were stained against 4-HNE, and levels of lipid peroxidation were monitored by flow cytometry. While conditions of low AO showed a marked increase in 4-HNE-staining, this was absent in Fer-1 treated organoids (Supplementary Fig. 2d). Together, these results show that omission of antioxidants during cortical neuronal differentiation increases ferroptotic cell death, which prevents proper neuronal differentiation. Specific inhibition of ferroptosis is, therefore, necessary for cortical neuron differentiation.

## Ferroptosis inhibition is critical for proper laminar organization of cortical organoids

We next investigated the influence of ferroptosis on the laminar organization of cortical organoids, a model of neurodevelopment with physiological relevance. At day 60, organoids generated without antioxidants (-AO + vA$^{lo}$) resulted in a diffused and unorganized distribution of CTIP2 + (Fig. 3a), SATB2 + (Fig. 3b) and TBR1 + (Fig. 3c) neurons, all markers for proper patterning[18]. In contrast, CTIP2 +, SATB2 + and TBR1 + neurons in the organoids were structured in the presence of antioxidants (+ AO) or ferrostatin-1 (-AO + vA$^{lo}$ + Fer-1), thereby generating a sharp CTIP2, TBR1 as well as SATB2 layer (Fig. 3a–c and Supplementary Fig. 3). Moreover, ferroptosis events in -AO + vA$^{lo}$ organoids also altered the well-defined ventricular-zone-like structure in forebrain organoids, which contain SOX2+ neural progenitor cells (Fig. 3c).

We further quantified the distribution of CTIP2 +, TBR1 +, SATB2 + and SOX2 + cells by dividing the developing cortex into five equal bins, as described previously[18,25], with bin 1 corresponding to ventricular zone (VZ), bins 2 and 3 corresponding to subventricular zone (SVZ), and bins 4 and 5 corresponding to the cortical plate (CP) of the cortex (Fig. 3a–c and Supplementary Fig. 3). Cells without antioxidants (-AO + vA$^{lo}$) undergoing ferroptosis showed significantly more CTIP2 + and SATB2 + cells localized in bins 1 to 4 and fewer cells localized in bin 5 compared to control cells treated with antioxidants (+ AO) or Fer-1 (-AO + vA$^{lo}$ + Fer-1) (Fig. 3a, b). The inverse distribution was observed for TBR1 + neurons; i.e., more cells in bins 3 to 5 in conditions without antioxidants (-AO + vA$^{lo}$) compared to control cells (+ AO) or Fer-1 treated cells (-AO + vA$^{lo}$+Fer-1) (Supplementary Fig. 3). The same observation was made for SOX2 + cells, which were shifted more toward bins 3–5 in antioxidant-negative conditions (Fig. 3c). Hence, our data demonstrate impaired laminar organization of cortical organoids when organoids were grown without antioxidants, and this phenotype was reversed when ferroptosis was specifically inhibited with ferrostatin-1, proving the importance to suppress ferroptosis during neuronal differentiation.

## Iron-overload-mediated growth defect during *C. elegans* development is ameliorated by ATRA

We next used an iron-overload in vivo model in *C. elegans* to test whether vitamin A (ATRA) has a ferroptosis-protective effect. It has previously been shown that treatment of *C. elegans* with ferric ammonium citrate (FAC) causes ferroptosis in vivo[26]. Accordingly, 72 h of FAC treatment during development significantly reduced worm growth, which we measured as length, compared to controls (Fig. 4a). In parallel, we treated worms for 72 h with FAC and vitamin E (Trolox) – a well-known ferroptosis inhibitor – as well as vitamin A (ATRA) to reduce ferroptosis-dependent growth defects, and monitored worm

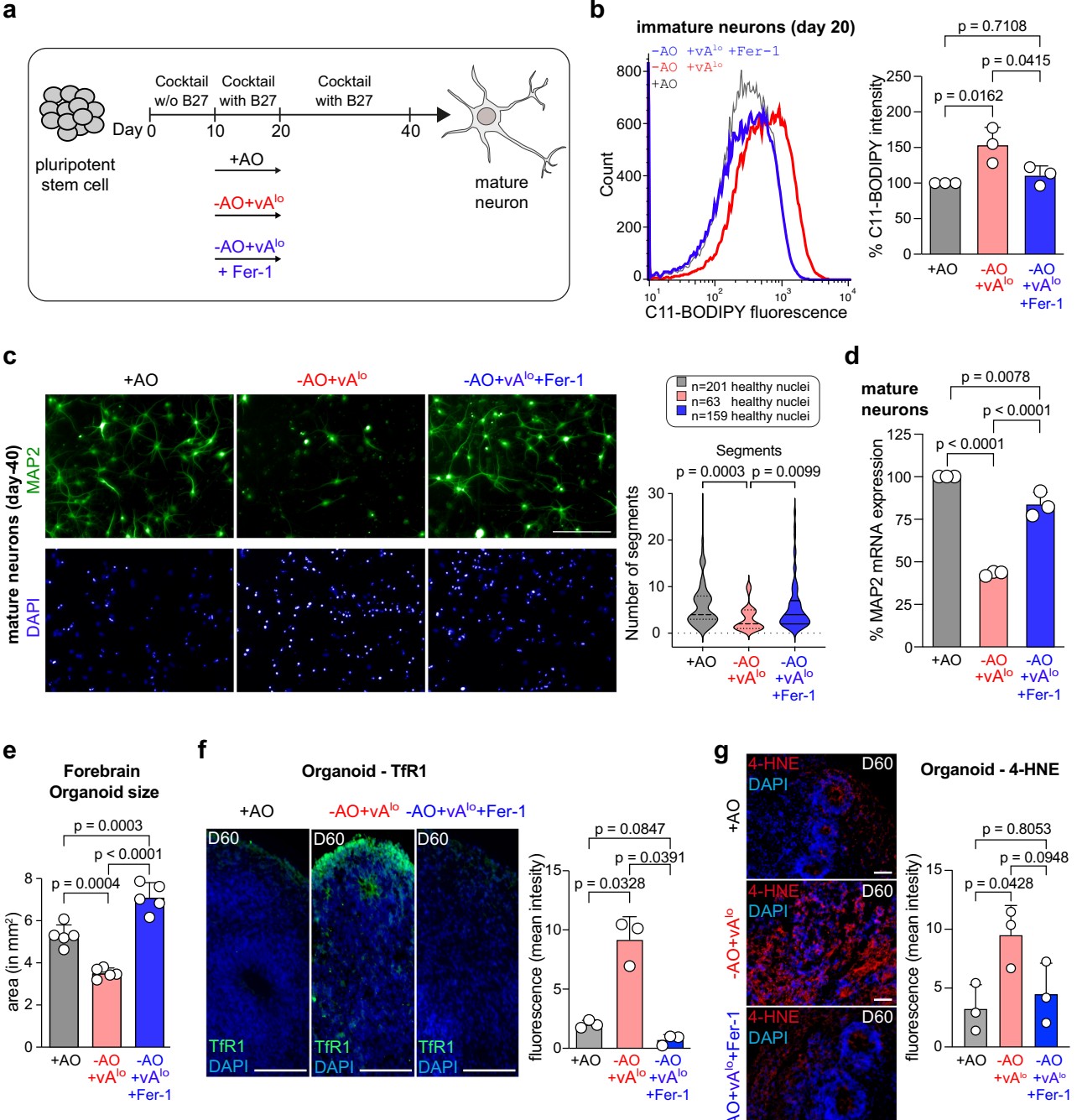

**Fig. 2 | Inhibition of ferroptosis by ferrostatin-1, upon deprivation of anti-oxidants, facilitates neuronal differentiation. a** Scheme of cortical neuronal differentiation using three conditions (i) standard media with antioxidant-containing B27 (+ AO), (ii) media with antioxidant-deficient B27 containing vitamin A (-AO + vA$^{lo}$), or (iii) media with antioxidant-deficient B27 containing vitamin A, which was further supplemented with ferrostatin-1 (Fer-1) (-AO + vA$^{lo}$ + Fer-1). **b** C11-BODIPY staining of day 20 immature cortical neurons generated with antioxidant (+ AO) or without antioxidants (-AO + vA$^{lo}$) or supplemented with ferrostatin-1 (-AO + vA$^{lo}$ + Fer1) using flow cytometry. (Left) flow cytometry histograms; (right) median intensity. Data are mean ± SD of $n = 3$ biologically independent replicates; one-way ANOVA with Tukey's test. **c** (Left), MAP2 immunofluorescence staining of cortical neurons at day 40 generated with antioxidant (+ AO) or without anti-oxidants (-AO + vA$^{lo}$) or supplemented with ferrostatin-1 (-AO + vA$^{lo}$ + Fer-1). (Right), high-content image analyses. $n = 3$ biologically independent replicates (with approx. 30–70 segments for individual samples); Kruskal-Wallis with Dunn's test. Scale bars, 100 μm. **d** MAP2 mRNA expression of cortical neurons generated with

antioxidant (+ AO) or without antioxidants (-AO + vA$^{lo}$) or supplemented with ferrostatin-1 (-AO + vA$^{lo}$ + Fer-1) using quantitative RT-PCR. Data are mean ± SD of $n = 3$ biologically independent replicates; one-way ANOVA with Tukey's test. **e** Area of organoids at day 60 generated with antioxidant (+ AO) or without antioxidants (-AO + vA$^{lo}$) or supplemented with ferrostatin-1 (-AO + vA$^{lo}$ + Fer-1) and measured by ImageJ. Data are mean ± SD of $n = 5$ independent organoids; one-way ANOVA with Tukey's test. **f** (Left), organoid sections stained for the ferroptosis marker TfR1 at day 60 of forebrain generation. (Right), measurement of the TfR1 fluorescence intensity using ImageJ. Data are mean ± SD of $n = 3$ biologically independent replicates; one-way-ANOVA with Tukey's test. Scale bars, 50 μm. **g** (Left), organoid sections stained for 4-HNE lipid peroxidation marker at day 60 of forebrain generation. (Right), measurement of the 4-HNE intensity using ImageJ. Data are mean ± SD of $n = 3$ biologically independent replicates with 5 image sections per replicate; one-way-ANOVA with Tukey's test. Scale bars, 50 μm. Source data are provided as a Source Data file.

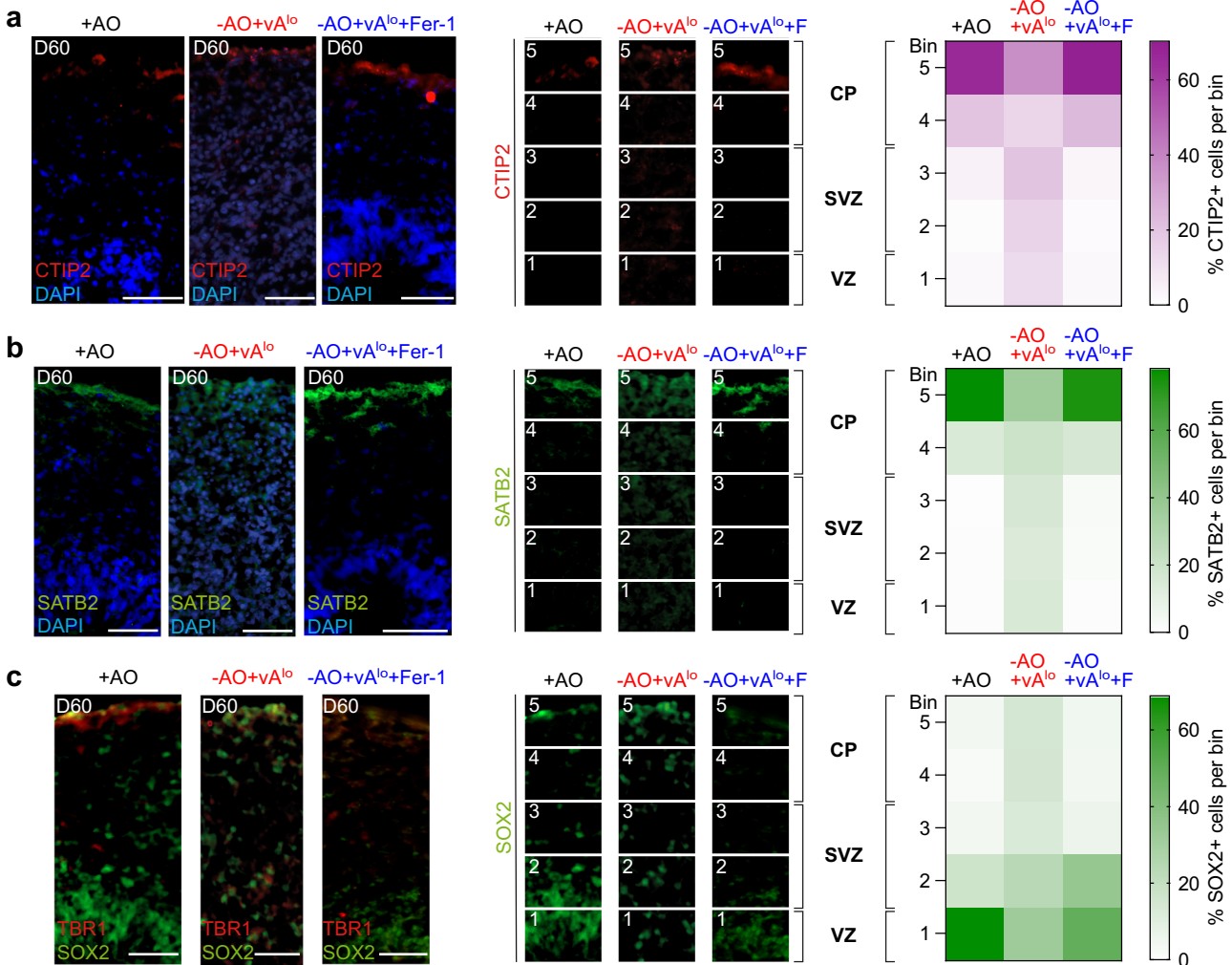

**Fig. 3 | Inhibition of ferroptosis is critical for correct laminar organization of cortical organoids. a–c** (Left), immunofluorescence of day 60 forebrain organoid sections generated with antioxidant (+ AO) or without antioxidants (-AO + vA$^{lo}$), or the latter supplemented with ferrostatin-1 (-AO + vA$^{lo}$ + Fer-1), and stained for (**a**) CTIP2/DAPI, (**b**) SATB2/DAPI, and (**c**) TBR1/SOX2. Scale bars, 50 μm. (Middle), quantification of CTIP2 + , SATB2 + and SOX2 + cells using ImageJ analysis. *n* = 3 biologically independent replicates. CP, cortical plate; SVZ, subventricular zone; VZ, ventricular zone. (Right), distribution of CTIP2 + , SATB2 + and SOX2 + cells illustrated using heatmaps; F = Fer-1. Source data are provided as a Source Data file.

length. Importantly, we found that Trolox and ATRA treatment significantly ameliorated the FAC-induced growth defects in developing worms (Fig. 4a).

**Vitamin A inhibits ferroptosis by reducing lipid peroxidation**
Next, we aimed to understand the mechanism by which vitamin A counteracts ferroptotic cell death, and whether the effect is specific to neurons. Hence, we used the fibrosarcoma cell line HT-1080, which is well-established to study ferroptosis[5,10,12]. In an ATP-based viability assay, cells were treated with the ferroptosis inducer (FIN) RSL3 and co-treated with vitamin A (ATRA). ATRA significantly rescued cells from ferroptotic cell death (Fig. 4b and Supplementary Fig. 4a, c). A similar effect was achieved when ferroptosis was induced by the system x$_c^-$ inhibitor IKE (Supplementary Fig. 4b). Moreover, the anti-ferroptotic impact of ATRA was similar to ferrostatin-1 (Fer-1), a gold standard ferroptosis inhibitor[5], although at a different concentration range (Supplementary Fig. 4a, b). These data show that ATRA also inhibits ferroptosis in other cell models.

To elucidate whether this ferroptosis-inhibiting effect of vitamin A can be ascribed to a possible direct antioxidative property, we performed several cell-free assays: first, oxidation of BODIPY 581/591 C11 was induced by 2,2′-Azobis(2-amidinopropane) dihydrochloride (AAPH), a free radical initiator, in a cell-free setting. The addition of the radical-trapping antioxidant Fer-1 could stop the oxidation reaction, whereas the addition of ATRA had no effect (Fig. 4c). In a second antioxidation assay, we initiated a radical reaction using 2,2-diphenyl-1-picrylhydrazyl (DPPH), and again Fer-1 acted as a radical-trapping antioxidant, but ATRA showed no activity (Supplementary Fig. 4d). These data, in the absence of the cellular environment, suggest that the ferroptosis-inhibiting effect of vitamin A is not caused by any direct antioxidative properties. All-trans retinoic acid (ATRA) activates the RAR nuclear receptors to initiate gene transcription, raising the possibility that vitamin A (ATRA) inhibits ferroptosis via transcriptional control. To test this hypothesis, we co-treated HT-1080 cells with ATRA and small molecule inhibitors of Retinoic Acid Receptor (RARi, AGN193109) or Retinoid X receptor (RXRi, HX531), respectively, and observed that both inhibitors re-sensitized cells to RSL3-mediated ferroptosis in the presence of ATRA (Supplementary Fig. 4e, f). We also cultured HT-1080 spheroids to validate these findings in a 3D model (Fig. 4d and Supplementary Fig. 5). We induced ferroptotic cell death using RSL3, which consequently prevented the formation of proper spheroids. Treatment with ATRA restored the round shape of the

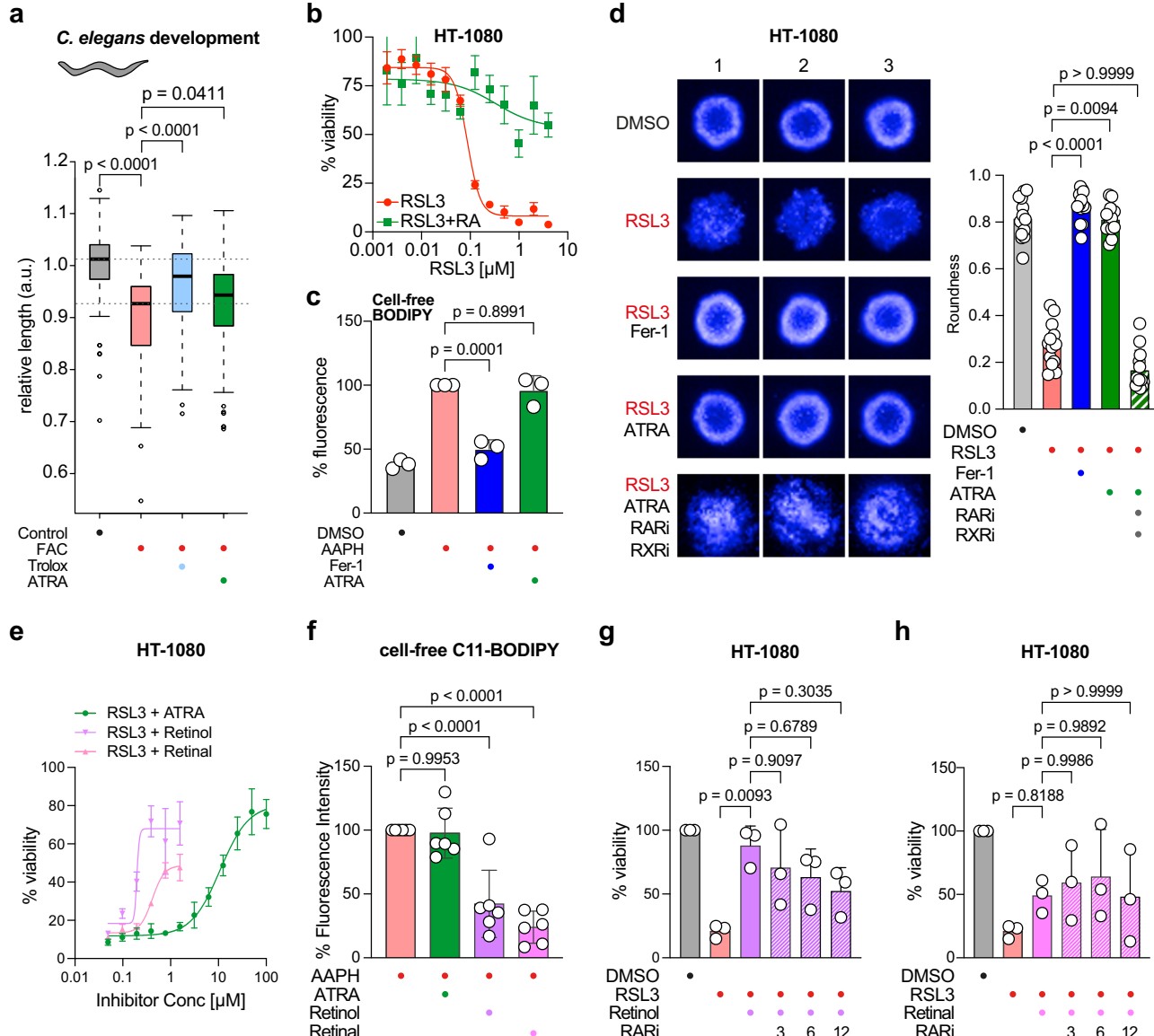

**Fig. 4 | Vitamin A reduces iron-overload-mediated developmental growth defect in *C. elegans* and suppresses ferroptosis in HT-1080 cells. a** Box plot showing the relative worm length at 72 h of development in control, FAC-treated or FAC-treated supplemented with Trolox or vitamin A animals. Data are from 4 biologically independent replicates (15–30 worms per biological replicate). The median is a thick line. Box limits are 25th and 75th percentiles, and whiskers denote 1.5 times the interquartile ranges. *p* values were calculated with a Wilcoxon rank sum test with continuity correction (two-sided). **b** CellTiter-Glo assay of HT-1080 cells co-treated with RSL3 and ATRA (*n* = 3 biologically independent replicates). Data shown are mean ± SD. **c** C11-BODIPY cell-free assay of the radical initiator 2,2′-Azobis(2-amidinopropane) dihydrochloride (AAPH) co-treated with Fer-1 or ATRA. Data are mean ± SD of *n* = 3 biologically independent replicates; one-way-ANOVA with Tukey's test. **d** (Left), HT-1080 spheroids co-treated with RSL3 and Fer-1, ATRA, or in combination with inhibitors of Retinoic Acid Receptor (RARi) and Retinoid X

receptor (RXRi). (*n* = 13; remaining spheroids in Supplementary Fig. 5) (Right), quantification of spheroid roundness. Data are mean ± SD of *n* = 13 independent spheroids; Kruskal-Wallis with Dunn's test. **e** CellTiter-Glo assay of HT-1080 cells co-treated with RSL3 and ATRA, Retinol or Retinal (*n* = 3 biologically independent replicates) – Retinal and Retinol at concentrations > 3 μM were toxic. Data shown are mean ± SD. **f** C11-BODIPY cell-free assay of the radical initiator AAPH co-treated with ATRA, Retinol or Retinal. Data are mean ± SD of *n* = 6 biologically independent replicates; one-way-ANOVA with Tukey's test. **g** CellTiter-Glo assay of HT-1080 cells after co-treatment with RSL3 and Retinol (2 μM) as well as different concentrations of RARi (μM). Data are mean ± SD of *n* = 3 biologically independent replicates; one-way-ANOVA with Tukey's test. **h** CellTiter-Glo assay of HT-1080 cells co-treated with RSL3 and Retinal (0.8 μM) as well as different concentrations of RARi (μM). Data are mean ± SD of *n* = 3 biologically independent replicates; one-way-ANOVA with Tukey's test. Source data are provided as a Source Data file.

spheroids comparable to Fer-1. Interestingly, co-treatment of ATRA with RARi/RXRi re-sensitized spheroids to RSL3 destruction, leading to loss of roundness as a result of ferroptotic cell death. These results were quantified by performing a Hoechst 33342 stain with subsequent high-content-image analysis that calculated spheroid roundness (Fig. 4d).

Vitamin A exists in several forms. ATRA, as used above, is the active metabolite that activates RAR, but precursor forms include

retinal and retinol[27]. Importantly, retinol can be converted to retinal, and retinal can be further converted to ATRA[27]. To understand the effect of additional vitamin A metabolites, we tested the ability of retinol and retinal to inhibit ferroptosis. Using cell viability assays, we can show that retinol and retinal suppress ferroptosis in a dose-dependent manner, with retinol being more potent than retinal (Fig. 4e). Notably, retinol and retinal are toxic to HT-1080 cells at concentrations above 3 μM. Next, we investigated whether retinol and

retinol have antioxidant capacity. Importantly, retinol and retinal markedly reduced the cell-free C11-BODIPY signal induced by AAPH (Fig. 4f); thus, demonstrating that these two vitamin A metabolites have radical-trapping antioxidant activity. We also verified that RAR inhibition had a lesser effect on ferroptosis inhibition by retinol and retinal (Fig. 4g, h). These data reveal that different vitamin A metabolites have different mechanisms of action, i.e., retinol and retinal via radical-trapping antioxidation and ATRA via transcriptional control by activating RAR.

Next, we investigated whether ATRA, although without direct antioxidative function, would be able to reduce lipid peroxidation in a series of cell-based lipid peroxidation assays. First, in a live cell staining with the fluorescent lipid peroxidation sensor C11-BODIPY 581/591 and subsequent flow cytometry, we observed that ATRA decreased RSL3-induced lipid peroxidation (Fig. 5a). We also performed C11-BODIPY 581/591 in microscopy imaging and found reduction of lipid peroxidation by ATRA (Supplementary Fig. 6a). Further, by immunostaining and flow cytometry, another RSL3-induced product of lipid peroxidation in cell membranes, namely 4-hydroxynonenal (4-HNE), was found to be significantly reduced upon ATRA treatment (Fig. 5b). Finally, we performed a TBARS assay that measures levels of malondialdehyde (MDA), a product of lipid peroxidation, and found that ATRA significantly reduced MDA levels similar to Fer-1 indicating a reduction of lipid peroxidation (Fig. 5c). Hence, these studies demonstrate that ATRA addition eliminates lipid peroxidation.

## ATRA upregulates major anti-ferroptotic regulators

Based on these findings that ATRA activates RAR/RXR to reduce lipid peroxidation, we hypothesized that ATRA may alter the expression of distinct ferroptosis-inhibiting regulators involved in the suppression of lipid peroxidation. To verify this, quantitative RT-PCR experiments were performed in HT-1080 cells treated with ATRA. Several genes that are established as key ferroptotic gatekeepers were significantly upregulated upon ATRA treatment, i.e., GPX4, FSP1, and GCH1—the three major pillars of anti-ferroptotic defense (Fig. 5d). In addition, expression of anti-ferroptotic regulators involved in lipid biosynthesis and lipid metabolism were increased: Acyl-CoA Synthetase Long-chain family member 3 (ACSL3), Stearoyl-CoA desaturase (SCD1), and Peroxisome Proliferator-Activated Receptor alpha (PPARα) (Fig. 5d). Importantly, when cells were co-treated with ATRA and RARi the expression of GPX4, FSP1, GCH1, ACSL3, SCD1, and PPARα was reduced to baseline levels, again proving that ATRA upregulates the ferroptosis suppressors through a RAR transcriptional control (Fig. 5d). We further co-treated HT-1080 cells with RSL3, ATRA and inhibitors against FSP1 (iFSP1) or PPARα (GW6471) (Supplementary Fig. 6b). Cotreatment with these inhibitors resulted in a loss of the anti-ferroptotic effect of ATRA to re-sensitize the cells towards ferroptosis; thus, proving these target genes are indeed downstream target of RAR/RXR activation. To prove that also protein levels of ferroptosis gatekeepers were upregulated, we checked the expression of GPX4, FSP1, ACSL3, and SCD1 by Western Blot. Protein levels of these cellular regulators were increased upon ATRA treatment (Fig. 5e). Interestingly, the protein expression of the pro-ferroptotic regulators ACSL4 and LPCAT3 were reduced upon ATRA treatment (Fig. 5e), which may be due to an indirect effect such as upregulation of an E3 ligase.

We finally evaluated whether this mechanism was also responsible for the anti-ferroptotic effect seen in immature neurons and performed quantitative RT-PCR of neurons at day 20 of differentiation, which were grown without antioxidants, and either supplemented with lower or higher levels of vitamin A (Fig. 5f). Only higher vitamin A levels upregulated GPX4, FSP1, GCH1, ACSL3, and SCD1 (Fig. 5d and Supplementary Fig. 6c), thereby explaining the protective effect of vitamin A (ATRA) during stem cell differentiation by reducing lipid peroxidation and ferroptosis (Fig. 1d). PPARα was also upregulated to some extent upon higher vitamin A treatment, but was not significant

(Supplementary Fig. 6c). Our initial RNAseq analysis showed a slight increase of GPX4 and FSP1 in conditions without AO and vA^lo (Supplementary Fig. 1b), which is not enough to overcome ferroptosis and facilitate neuronal differentiation (Fig. 2c). Only higher levels of ATRA led to robust expression of anti-ferroptotic genes (Fig. 5d)

Thus, we report an unexpected function of vitamin A and its active component ATRA in inhibiting lipid peroxidation by enhancing the transcription of essential anti-ferroptotic defence regulators, including the main gatekeepers GPX4, FSP1 and GCH1, as well as the lipid composition regulators ACSL3, SCD1, and PPARα. Moreover, our work uncovers that the process of early neuronal development requires the suppression of ferroptosis by antioxidants (e.g., vitamin E or ferrostatin-1) or vitamin A to generate mature neurons.

## Discussion

In this study, we found that overcoming ferroptosis is a critical requirement for the differentiation of pluripotent stem cells into neurons. We demonstrate that ferroptosis can be inhibited by the use of either antioxidants (GSH and vitamin E) or ferrostatin-1, and we have identified vitamin A as a novel ferroptosis suppressor (model in Fig. 6). Interestingly, protocols for stem cell differentiation mostly contain antioxidants[16,28–33], but the exact mechanism of why these antioxidants are essential has been unclear. The differentiation process uses transferrin and PUFAs (Supplementary Data 1), both of which are involved in promoting lipid peroxidation. In vivo, ROS levels are important for proliferation and differentiation events during early mouse embryonic brain development, but continuous increase of ROS leads to oxidative stress and lipid peroxidation[34]. Therefore, a tight control is needed. Interestingly, NRF2, a master regulator of oxidative stress, has been shown to have limited expression in neurons throughout the central nervous system (CNS)[35]. Accordingly, key NRF2 target genes such as SLC7A11 also showed reduced expression in the CNS[35]. Hence, additional antioxidative systems need to be present to control ROS-mediated oxidative stress. Our work resolves that during the early phases of cortical neuron differentiation, lipid peroxidation and ferroptosis need to be prevented by antioxidants (e.g., vitamin E) to enable neuronal development. Notably, ferroptotic cell death affected neural progenitor (radial glia) cells and/or the maturing neurons. From our organoid data, we propose that progenitor cells undergo ferroptosis as cells are 4-HNE-positive and fewer TBR1+ cells reach the upper layer (cortical plate), leading to enrichment of cells in the VZ and SVZ. Future research will decipher the exact cell types affected by ferroptosis during neurogenesis by applying single-cell and spatial transcriptomics. Physiologically, a recent study showed that vitamin E is necessary for zebrafish nervous system development[36]. Also, vitamin E deficiency leads to the accumulation of lipid peroxides in the zebrafish nervous system and impairs cognitive functions[37], which supports our findings. Importantly, we further demonstrate that ATRA can ameliorate iron-overload-induced growth defects in vivo during C. elegans development. Notably, our discovery of inhibiting ferroptosis during early neuronal development may be generalizable to developmental processes: ROS levels are increased during mammalian reproduction from fertilized egg to implantation, where ROS is needed for signaling purposes and embryo development; however, elevated levels of ROS can be lethal[38]. Therefore, the female reproductive tract is enriched with vitamin E, vitamin A and GSH[38] which are able to directly or indirectly act as antioxidants to control ROS levels.

Several previous studies have described that vitamin A plays a critical role in various aspects of neuronal development by regulating gene expression, cell differentiation, neurite outgrowth, synaptogenesis, and neurotransmitter synthesis via its active metabolite ATRA, which has been described in neuronal cell models as well as in vivo studies[39–41]. ATRA regulates the expression of genes involved in neuronal fate determination and patterning of the nervous system[40]. This influence is based on the ability to activate the nuclear receptors

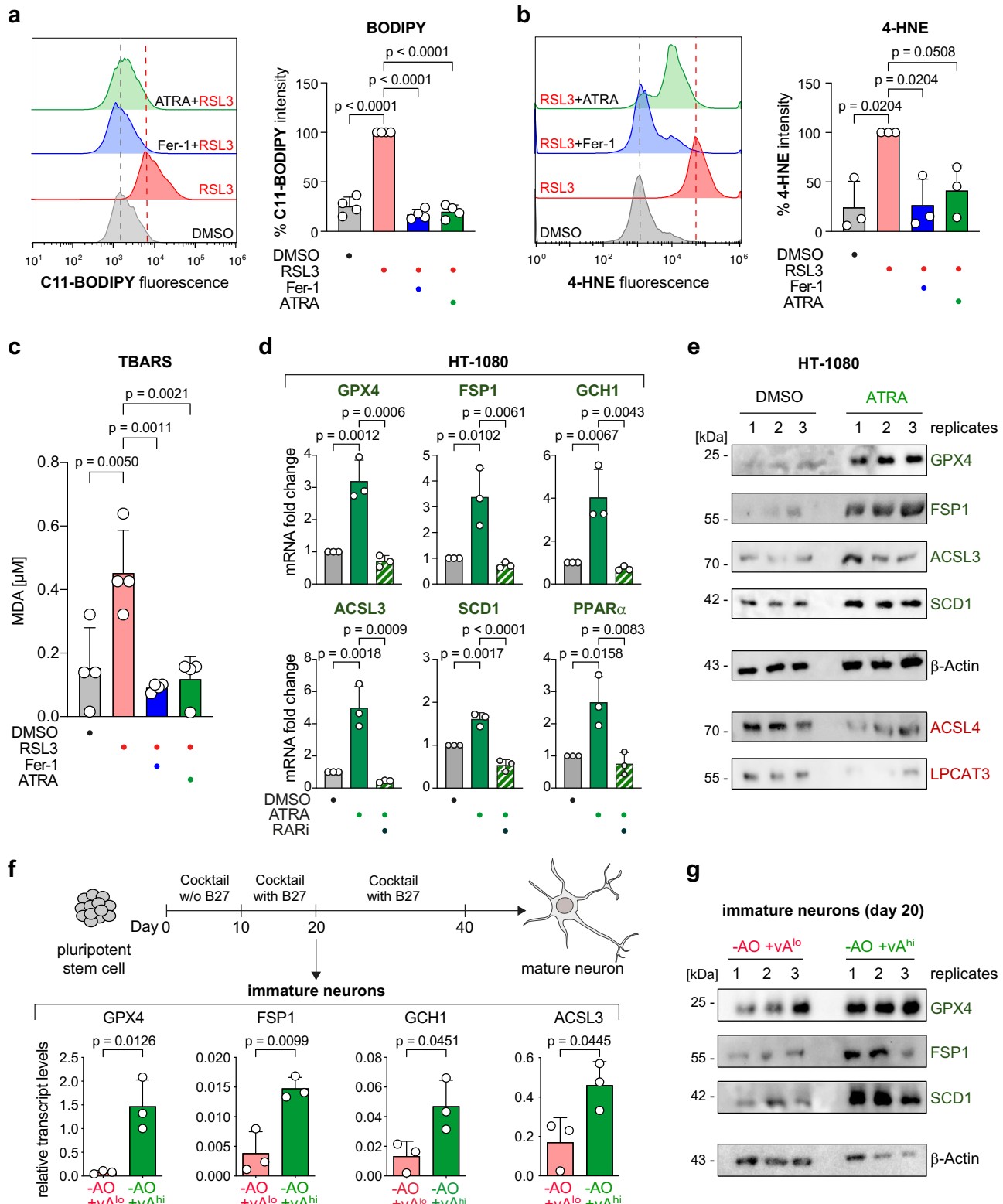

RAR/RXR for transcriptional induction of important developmental genes[21]. In mice, retinoic acid is expressed very early during neuronal development and activates a series of important genes that control brain patterning and progenitor differentiation[42]. At the same time, there is evidence that vitamin A has antioxidant capacity[43]. While there has been long-lasting debate about a direct or indirect antioxidant effect of vitamin A, there is now a consensus understanding that ATRA indirectly acts as an antioxidant by transcriptionally regulating

antioxidative mediators[43]. Two recent studies, however, described that metabolites of vitamin A have radical-trapping activity to inhibit ferroptosis[44,45], although ATRA seemed to have a weaker effect. In contrast to the previous studies, we unravel a dual mode of action of vitamin A in ferroptosis regulation: retinal and retinol have antioxidant properties, while ATRA activates RAR/RXR to transcriptionally upregulate GPX4, FSP1, GCH1, ACSL3, among other genes that are key gatekeepers to eliminate lipid hydroperoxides and to inhibit

**Fig. 5 | Vitamin A eliminates lipid peroxidation and upregulates major anti-ferroptotic regulators in HT-1080 as well as in cortical neurons. a** Flow cytometry of C11-BODIPY staining: (left) histograms and (right) intensity of HT-1080 cells co-treated with the ferroptosis inducer (RSL3), and inhibitor (Fer-1) or vitamin A (ATRA). Data are % intensity of median fluorescence normalized to RSL3-treated cells ± SD of $n = 4$ biologically independent replicates; one-way-ANOVA with Tukey's test. **b** Immunostaining of 4-Hydroxynonenal (4-HNE) and flow cytometry: (left) histograms and (right) intensity of HT-1080 cells co-treated with the ferroptosis inducer (RSL3), and inhibitor (Fer-1) or vitamin A (ATRA). Data are % intensity of median fluorescence normalized to RSL3-treated cells ± SD of $n = 3$ biologically independent replicates; one-way-ANOVA with Tukey's test. **c** TBARS assay of HT-1080 cells co-treated with the ferroptosis inducer (RSL3) and inhibitor (Fer-1) or vitamin A (ATRA). Data are mean ± SD of $n = 4$ biologically independent replicates; one-way-ANOVA with Tukey's test. **d** mRNA expression, using quantitative RT-PCR,

of various anti-ferroptotic regulators in HT-1080 cells treated with DMSO or 10 μM vitamin A (ATRA) and 10 μM RARi. Data are mean ± SD of $n = 3$ biologically independent replicates; one-way-ANOVA with Tukey's test. **e** protein expression, using Western Blot, of various anti-ferroptotic regulators (GPX4, FSP1, ACSL3, SCD1) as well as pro-ferroptotic regulators (ACSL4, LPCAT3) in HT-1080 cells treated with DMSO or 20 μM vitamin A (ATRA). $n = 3$ biologically independent replicates are shown. **f** mRNA expression (qRT-PCR) of various anti-ferroptotic regulators in cortical neurons differentiated with low or high vitamin A. Data plotted are mean ± SD of $n = 3$ biologically independent replicates; unpaired $t$ test, two-tailed. **g** protein expression, using Western Blot, of various anti-ferroptotic regulators (GPX4, FSP1, SCD1) in immature neurons differentiated with low or high vitamin A. $n = 3$ biologically independent replicates are shown. Source data are provided as a Source Data file.

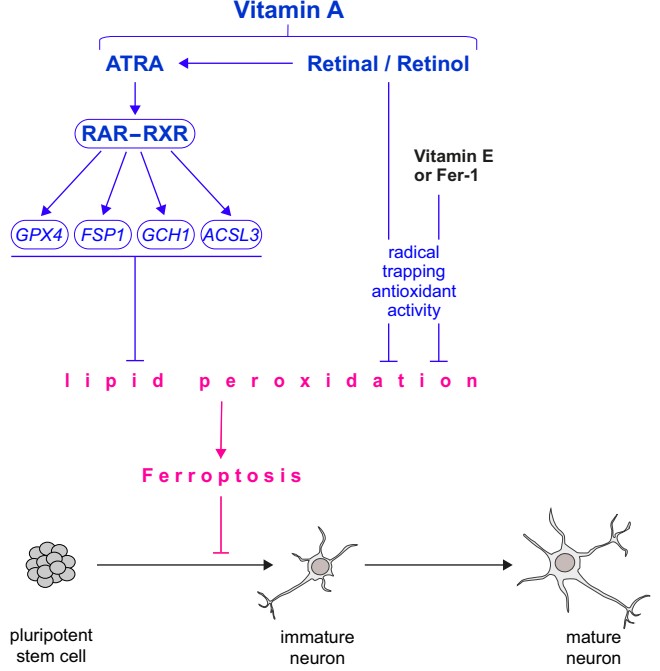

**Fig. 6 | Model of vitamin A or radical-trapping antioxidants suppressing ferroptosis to promote neuronal development.** This graphical summary shows that lipid peroxidation and ferroptosis must be suppressed to generate mature neurons. This is achieved either through the action of radical-trapping antioxidants (RTAs) or by transcriptional up-regulation of ferroptosis gatekeepers GPX4, FSP1, GCH1, and ACSL3 by all-trans retinoic acid (vitamin A). Vitamin A has a dual function in suppressing ferroptosis as both retinal and retinol have RTA activity.

ferroptosis. This dual mechanism of vitamin A – antioxidant property (retinal and retinol) and transcriptional rewiring (ATRA) – to counteract ferroptosis is distinct from vitamins E and K[5,11,12], as vitamins E and K act solely through radical-trapping antioxidant activity.

Together, our study answers a substantial question of why antioxidants are needed in early neurogenesis: overcoming ferroptotic cell death during early neurogenesis is essential to facilitate neuronal development. Moreover, this report establishes vitamin A as a potent ferroptosis inhibitor with a distinct dual antioxidant mechanism.

## Methods
### Chemicals
For induction of ferroptotic cell death, we used (1S,3 R)-RSL3 (Sigma-Aldrich) and Imidazole Ketone Erastin (IKE, Cayman Chemical). As ferroptosis inhibitors, we used ferrostatin-1 (Fer-1, Sigma-Aldrich), all-

trans-retinoic acid (ATRA, Sigma-Aldrich), all-trans-retinol (Sigma-Aldrich) and all-trans-retinal (MedChemExpress).

For inhibition of different receptors and proteins, we used HX 531 (Biomol) as a pan-retinoic Acid Receptor (RAR) antagonist, AGN 193109 (Sigma-Aldrich) as a Retinoic X Receptor (RXR) antagonist, iFSP1 (MedChemExpress) as an inhibitor of Ferroptosis Suppressor Protein 1 (FSP1) and GW6471 (Selleck Chemicals) as a PPARα antagonist.

### Cell culture
HT-1080: The fibrosarcoma cell line HT-1080 (purchased from ATCC) was cultured in Dulbecco's Modified Eagle's Medium supplemented with 10% fetal bovine serum, 1% non-essential amino acids (MEM NEAA) and 1% Penicillin-Streptomycin (all purchased from Thermo Fisher Scientific). The cell line was grown at 37 °C and 5% $CO_2$. HT-1080 was not further authenticated.

Human stem cells: H9 (WA09) cells were maintained with Essential 8 Flex media (#A28558501, Thermo Fisher Scientific) in feeder-free conditions on vitronectin (VTN-N) (#A14700, Thermo Fisher Scientific). H9 (WA09) cells were passaged as clumps with 0.5 M EDTA (0.5 M EDTA, 5 M NaCl, 1X PBS). The pluripotency level of the cells in culture was routinely authenticated for the markers Nanog and Oct-4.

Both cell lines were regularly checked for mycoplasma contamination via PCR.

### Cell viability assay
To test cell viability or create dose-response curves, 750 HT-1080 cells per well were seeded into 384-well plates (CulturPlate, PerkinElmer) and incubated for 24 h. Cells were treated with indicated RSL3 concentrations and 20 μM ATRA for 18 h; for dose-response curves, compounds were diluted in a 12-point series. Experiments with small molecule inhibitors were performed with the following concentrations: 200 nM RSL3, 1,5 μM IKE, 20 μM ATRA, and indicated concentrations of RARi, RXRi, iFSP1, or PPARαi, respectively. After treatment for 18 h, 20 μl CellTiter-Glo 2.0 Reagent (Promega) was added directly into each well, and luminescence was detected using an EnVision 2104 Multilabel plate reader (PerkinElmer).

### Spheroid formation, treatment and imaging
Spheroids were cultured according to the protocol previously described[15]. In brief, HT-1080 were seeded into 96-well round-bottom microplates (CellCarrier Spheroid ULA, PerkinElmer) with a density of 2000 cells per well. After 48 h of proliferation, spheroids were treated with 200 nM RSL3, 20 μM ATRA, 2 μM Fer-1, 10 μM RARi, and 10 μM RXRi, and incubated another 48 h. Finally, spheroids were stained with a 1:10,000 dilution of Hoechst 33342 (Sigma) and after 1 h of incubation at 37 °C, imaging was performed using an Operetta high-content system. For image analysis, the Columbus software (PerkinElmer) was used.

## 4-HNE staining and flow cytometry

Immunostaining of HT-1080 with anti-4-Hydroxynonenal-antibody (4-HNE, ab46545, Abcam) was performed as previously described[15]. In brief, ferroptotic cell death was induced via 300 nM RSL3 for 2 h, and cells were co-treated with 20 μM ATRA or 2 μM ferrostatin-1. 10% normal goat serum (Thermo Fisher Scientific) was used for blocking before cells were incubated in anti-4-HNE antibody. As a secondary antibody, an anti-rabbit Alexa 488 antibody (A32731, Thermo Fisher Scientific) was used. For flow cytometry, 10,000 events per condition were measured in an Attune acoustic flow cytometer (Applied Biosystems). FlowJo v10.8.1 Software (BD Life Sciences) was used to create histograms and analyze intensities. Differentiated neurons and dissociated forebrain organoids were stained, measured and analyzed with the same method.

For performing flow cytometry analysis of single cells from 3D forebrain organoids using 4-HNE staining, matured organoids from day 40 were used. Dissociation of the organoids was done by adding 1 mL of Accutase (Gibco; A1110501) into a 1.5 ml microcentrifuge tube containing the organoids and incubated at 37 °C for 30 min. The dissociated cells were pelleted by centrifuging at $500 \times g$ for 5 min at 4 °C and washed with 1 ml of 1x PBS at $500 \times g$ for 5 min at 4 °C. 1 ml of 70% ice-cold ethanol was added and incubated for 1 h on ice followed by centrifugation at $500 \times g$ for 15 min at 4 °C. The supernatant was discarded, and the cells were incubated in 2 drops of 10% normal goat serum (NGS, Invitrogen; #31872, RRID: AB_2532166) for 30 min on ice. 100 μl of primary antibody dilution (Rabbit polyclonal anti-4HNE, Abcam ab46545 in 1% BSA in 1x PBS, 1:50) was added and incubated for 30 min at room temperature. The cells were then washed with antibody dilution buffer (1% BSA in 1x PBS) three times at $500 \times g$ for 5 min, incubated with secondary antibody (AlexaFluor Goat anti-Rabbit 488 Thermo Fischer Scientific A32731) and washed for 3 times. The cells were then resuspended in 300 μl of antibody dilution buffer and proceeded for flow cytometry analysis using an Attune acoustic flow cytometer (Applied Biosystems). An unstained control and a secondary antibody control were maintained in the experimental setup. FlowJo v10.8.1 Software (BD Life Sciences) was used to create histograms and analyze intensities.

## C11-BODIPY staining

Microscopy: HT-1080 cells were seeded into 6-well plates with a density of 400,000 cells per well and incubated for 24 h, then 1 μM C11-BODIPY 581/591 (Thermo Fisher) was added for 30 min. Without changing the medium, cells were then treated with 250 nM RSL3 and 20 μM ATRA. After 2 h of treatment, the cell culture medium was replaced with PBS, and fluorescence was detected in the GFP channel of an EVOS FL fluorescence microscope (Thermo Fisher Scientific) using 20-fold magnification.

Flow cytometry: C11-BODIPY staining and flow cytometry were performed as previously described[15]. Ferroptosis was induced for 2 h with 250 nM RSL3 and cells were co-treated with 20 μM ATRA or 2 μM ferrostatin-1. HT-1080 cells were stained with 2 μM C11-BODIPY for 30 min. Day-20 neurons were stained with 2 μM C11-BODIPY for 1 h. Afterwards cells were measured in an Attune acoustic flow cytometer (Applied Biosystems). FlowJo v10.8.1 Software (BD Life Sciences) was used to create histograms and analyze intensities.

## TBARS Assay

HT-1080 cells were seeded into 100 mm dishes with a density of 2.5 million cells per dish and incubated for 48 h. Cells were then treated with 250 nM RSL3 for 3 h and co-treated with 20 μM ATRA or 2 μM ferrostatin-1 before they were harvested using 0.05% Trypsin-EDTA (Thermo Fisher Scientific) and cell scraper (Sarstedt). Cell counts of every treatment condition were determined and adjusted to the condition with the lowest count. TBARS (TCA Method) Assay Kit (Cayman Chemical, 700870) was performed according to the manufacturer's

instructions to detect levels of Malondialdehyde (MDA). Because the fluorometric method was chosen for detection, sample fluorescence was measured at 530 nm / 550 nm in an EnVision 2104 Multilabel plate reader (PerkinElmer).

## RNA isolation and quantitative RT-PCR

HT-1080: After treatment of HT-1080 cells with 10 μM ATRA and 10 μM RARi for 5 h, total RNA isolation was performed by using the Monarch Total RNA Miniprep Kit (New England BioLabs), and RNA concentration was detected in a NanoDrop 2000 spectrophotometer (Thermo Fisher Scientific). For the synthesis of complementary DNA, we used the Maxima H Minus First Strand cDNA Synthesis Kit, including DNA digestion (Thermo Fisher Scientific) oligo $(dT)_{18}$ primer and random hexamer primer. Quantitative RT-PCR was performed with PowerUp SYBR Green Master Mix (Thermo Fisher Scientific) in a LightCycler480 (Roche) using the respective primers listed in Supplementary Data 2. Quantification of gene expression was done using the $2^{(-\Delta\Delta Ct)}$ method[46], and obtained expression levels were normalized to RNA polymerase II expression.

Cortical neurons: Total RNA from cortical neurons was isolated using TRIzoL (#15596-026, Invitrogen) following the manufacturer's protocol. RNA extraction was performed using chloroform. RNA was precipitated in isopropanol and resuspended in nuclease-free water. Total RNA (500 ng) was reverse transcribed using the Superscript First Strand Synthesis kit following the manufacturer's protocol and random hexamers after DNAse treatment. qRT-PCR was performed using specific primers and the LightCycler 480 SYBR Green I (#04887352001, Roche) on a LightCycler 480 (Roche). Relative gene expression was calculated using the $2^{(-\Delta Ct)}$ method[47]. All genes were normalized to RNA polymerase II values.

Primer details are provided in Supplementary Data 2.

## SDS-PAGE and Western blotting

HT-1080 were seeded in a 6-well plate with a density of 200,000 cells per well, incubated for 24 h, and afterward treated with 20 μM ATRA for 6 h. Cells were harvested with 2X SDS buffer (Roti-Load, Roth), collected via scraping, and sonicated using a Sonifier U9200S (Hielscher Ultrasonics). For protein denaturation, samples were heated at 95 °C for 5 min.

Cell lysates were run on a NuPAGE™ 4–12% Bis-Tris gel (Invitrogen) in 1X MOPS SDS Running. Buffer (Invitrogen) and proteins were blotted onto a PVDF membrane using a semi-dry blotting system. The membrane was blocked with 5% milk in TBS-Tween and afterward incubated overnight at 4 °C in the following primary antibodies: rabbit anti-GPX4 (ab125066, Abcam, 1:1000), rabbit anti-FSP1 (AMID, PA5-103183, Thermo Fisher Scientific, 1:500), rabbit anti-ACSL3 (ab151959, Abcam, 1:1000), rabbit anti-SCD1 (ab236868, Abcam, 1:1000), mouse anti-ACSL4 (sc-365230, Santa Cruz Biotechnology, 1:500), mouse anti-LPCAT3 (ab239585, Abcam, 1:1000). For normalization, membranes were incubated in mouse anti-β-Actin (C4) (sc-47778, Santa Cruz Biotechnology, 1:500). Antibodies were diluted in 2.5% milk in TBS-Tween. After 3 washing steps in TBS-Tween, membranes were incubated for 1 h at room temperature in secondary antibody (goat anti-rabbit IgG (H + L) HRP conjugate, W4011; goat anti-mouse IgG (H + L) HRP conjugate, W4021, Promega), which was diluted 1:2500 in 2.5% milk in TBS-Tween. Chemiluminescence was detected using the Western Lightning ECL Pro Kit (PerkinElmer) and a Chemostar ECL imaging system (Intas). Full scans of Western Blots are shown in Supplementary Fig. 7.

## Cell-free C11-BODIPY and DPPH assays

For the cell-free C11-BODIPY assay, ATRA was diluted in 150 μl PBS to a final concentration of 25 μM. C11-BODIPY 581/591 (Thermo Fisher Scientific) was diluted to 1.875 μM and 2,2'-Azobis(2-methylpropionamidine) dihydrochloride (AAPH, Sigma) was diluted to 7.5 mM, each in

separate tubes of 150 μl PBS. All three tubes were mixed, vortexed, and incubated for 30 min at room temperature and protected from light. Triplicates of 100 μl were transferred into a black 96-well plate (Greiner Bio-One) and fluorescence (495 nm/520 nm) was measured with an EnVision 2104 Multilabel plate reader (PerkinElmer). As control samples, the same procedure was performed with ferrostatin-1 (25 μM) as a positive control, DMSO as a negative control, and one sample without the radical-inducing AAPH.

The 2,2-diphenyl-1-picrylhydrazyl (DPPH) Assay was performed according to a published protocol[5]. In brief, ATRA was diluted in DMSO to a final concentration of 10 mM. 5 μl of this dilution was added to 1 ml of 0.05 mM DPPH (Sigma, D9132) in methanol and rotated at room temperature for 10 min. 200 μl were transferred into a clear bottom 96-well plate (Corning costar) in quadruplicates and absorbance was measured at 517 nm with an EnVision 2104 Multilabel plate reader (PerkinElmer). As control samples, ferrostatin-1 and DMSO were measured with the same procedure. Background absorbance of methanol-only was subtracted from measured values.

## Differentiation of stem cells into cortical neurons

Coating plates for differentiation: Prior to the start of differentiation experiments, 24-well cell culture plates were coated with Geltrex (#A14133-02, Thermo Fisher Scientific) as previously described[18]. Briefly, Geltrex was thawed overnight on ice at 4 °C. The next day Geltrex was diluted in ice-cold DMEM/F12 (#11330-032/ Thermo Fisher Scientific) (ratio 1:30) and added to the plates (0.5 ml/well for 24-well plate or 2 ml/well for 6-well plate). Plates were sealed with parafilm and stored overnight at 4 °C.

Coating plates for replating with PO/Lam/FN: Before replating, poly-L-ornithine hydrobromide (PO) (#P3655/Sigma) was diluted to 15 mg/ml in PBS. 0.5 ml/well for a 24-well plate was added and incubated overnight at 37 °C/5% $CO_2$. The Next day, the PO solution was removed, and the plates were washed 3 times with 1x PBS. Afterward, Laminin I (3400-010-1, R&D systems) and Fibronectin (FC010, EMD Millipore Corp.) were diluted at 2 mg/ml in PBS and added at 0.5 ml/well to a 24-well plate. The plates were incubated overnight at 37 °C.

Differentiation into cortical neurons: The cortical neuron differentiations were carried out as previously described[16–18]. Briefly, human pluripotent stem cells were dissociated using Accutase and plated at high density on Geltrex (#A14133-02, Thermo Fisher Scientific) coated wells. The next day, the plates were checked for complete confluency and directed toward cortical neuron patterning. Cells were cultured in Essential 6 medium (E6, #A1516401, Thermo Fisher Scientific) in the presence of LDN193189 (#72142, Stem Cell Technologies), SB431542 (#1614, Tocris) and XAV939 (#3748, Tocris) (until day 5). From day 5 to 10, cells were cultured in the presence of TGFβ and BMP inhibitors (LDN193189, #72142, Stem Cell Technologies; SB431542, #1614, Tocris) to trigger cortical precursors for cortical neuron differentiation. From day 10 to 20, the cells were cultured in Neurobasal media (#21103-049, ThermoFisher Scientific) and DMEM F-12 media (#11330-032/ Thermo Fisher Scientific) supplemented with L-Glutamine (#25030-024, Thermo Fisher Scientific), B27(-vitamin A) (#12587010, Thermo Fisher Scientific), N2 supplement B (# 07156, StemCell Technologies), D-(+) Glucose, 2-mercaptoethanol, Sodium bicarbonate and Progesterone. From day 20 cortical neurons were cultured on polyornithine (PO)/ laminin (L)/fibronectin (FN) coated wells and kept in neuronal differentiation media (Neurobasal media (#21103-049, Thermo Fisher Scientific), 1% Pen/Strep (#15140-122, Thermo Fisher Scientific), L-Glutamine, B27(-vitamin A) supplemented with DAPT (#2634, Tocris) until day 30. Starting from day 30, DAPT was withdrawn and supplemented with BDNF (#450-10, PeproTech), GDNF (#248-BD-025, R&D Biosystems), cAMP (#D0627, Sigma) and AA (#4034-100, Sigma)) as described previously[16,18,31,48].

## Immunofluorescence of cortical neurons

At day 20, neurons were washed with 1X PBS and fixed with 4% PFA for 15 min at room temperature. Cells were permeabilized with 0.03% Triton X-100/PBS for 10 min and washed twice in 0.15% Triton X-100/ PBS at room temperature. Blocking was done in 0.15% Triton X-100/PBS supplied with 5% BSA (40 mg/mL) for 60 min at room temperature. Cells were incubated with the respective primary antibody (Mouse monoclonal Anti-MAP2 Sigma-Aldrich M-1406, RRI-D:AB_477171) in 0.15% Triton X-100/PBS supplied with 5% BSA (40 mg/ mL) overnight at 4 °C. Next day, cells were washed three times with 0.15% Triton X-100/PBS and incubated with the respective secondary antibody (AlexaFluor Goat Anti-Mouse 488 Thermo Fischer Scientific R37120, RRID:AB_2556548) in 0.15% Triton X-100/PBS supplied with 5% BSA for 2 h. Nuclear stain DAPI was added at a concentration of 1:10,000 90 min post-incubation with a secondary antibody. Finally, cells were washed three times with 0.15% Triton X-100/PBS and imaged using a Zeiss (AX10) or Zeiss LSM980 microscope.

## RNAseq and analysis in immature cortical neurons

Human stem cells were differentiated into cortical neurons. Between days 10 and 20, during cortical differentiation, cells were treated with a B27 supplement (+ AO) or with a B27 supplement without antioxidants (-AO), respectively. At day 20 of cortical neuronal differentiation total RNA sequencing (RNAseq) was performed. Therefore, total RNA from cortical neurons was isolated using TRIzoL (#15596-026, Invitrogen) following the manufacturer's protocol. The RNA integrity number (RIN) was measured using the Agilent 2100 Bioanalyzer. RNA with an RIN greater than 8 was sent for whole-genome RNA sequencing. The TruSeq RNA Sample Preparation Kit (Illumina) was used to synthesize and sequence the cDNA libraries. RNA libraries were prepared for sequencing using standard Illumina protocol. Sequencing was done in paired-end-reads using the HiSeq4000 sequencer (Illumina) at Helmholtz Zentrum Munich RNaseq Core Facility. The RNAseq data (count data) are provided in Supplementary Data 3.

Differential expression (DE) analysis was implemented with DESeq2 R package[49] to raw counts of RNA-seq data with drug replicates. Significance was assessed using the Wald Test, and p-values were adjusted for multiple hypothesis testing by the Benjamini-Hochberg procedure. Genes were defined as statistically differentially expressed with False Discovery Rate (FDR) < 5% and absolute value of $\log_2$ fold-change ($\log_2 FC$) > 1. All DE analysis and visualizations were performed in R software version 4.2.0 (R Foundation for Statistical Computing, https://www.r-project.org), and are available on GitHub (https:// github.com/MendenLab/DEx_StemCells_new).

## Generation of forebrain organoids

Forebrain organoids were generated from human pluripotent stem cells as described in refs. 18,50. Briefly, H9 cells were dissociated in Accutase. 9000 cells/well were seeded in a V-bottom ultra-low adhesion 96 well plate (#MS-9096 VZ, Sbio prime surface 96 V plate) in Essential 8 flex medium + Y-drug (#1254 Y-27632 dihydrochloride, Tocris), centrifuged at 3000 rpm for 10 min. From D0-D5, cells were cultured in Essential 6 medium (E6, #A1516401, Thermo Fisher Scientific) in the presence of LDN193189 (#04-0074, Stemgent), SB431542 (#1614, Tocris) and XAV-939 (#3748, Tocris). XAV-939 was removed from day 5 till day 18. From D18 onwards, organoids were maintained in organoid differentiation medium on an orbital shaker (50% DMEM F-12 (#11320-033, Thermo Fisher Scientific), 50% Neurobasal media (#21103-049, Thermo Fisher Scientific), 0.5x N2 supplement (#17502-048, Thermo Fisher Scientific), 0.025% insulin (#I-034, Sigma), 5 mM L-Glutamine (#25030024, Thermo Fisher Scientific), 0.7 mM MEM-NEAA (#11140050, Thermo Fisher Scientific), 50 U/mL Penicillin-Streptomycin (#15140-122, Thermo Fisher Scientific), 55 mM 2-mercaptoethanol (#21985-023, Thermo Fisher Scientific), 1xB27 supplement -vitamin A (#12587010, Thermo Fisher Scientific).

## Sectioning and immunostaining of forebrain organoids

For visualizing organoids, cells were fixed in 4% PFA overnight at 4 °C and washed three times in 1X PBS. After fixation, organoids were cryoprotected in 30% sucrose/PBS on a rotor shaker overnight at 4 °C and sectioned at 20 μm on a cryostat (Leica 1850 UV). Sections were permeabilized in 0,3% Triton X-100/PBS, blocked in 0.15% Triton X-100/PBS supplied with 5% BSA (40 mg/mL) and incubated as floating sections in primary antibody (Mouse monoclonal Anti-SOX2 Santa-Cruz Biotech sc-365823; Rat monoclonal Anti-CTIP2 Abcam ab18465; Rabbit polyclonal Anti-TBR1 Abcam ab31940; Rabbit polyclonal Anti-SATB2 Sigma-Aldrich HPA001042; Mouse Anti-TfR1 (CD71−3B8 2A1) antibody Santa Cruz sc-32272; rabbit anti-4-HNE, ab46545, Abcam) overnight. The next day, sections were washed three times with PBS and incubated with a secondary antibody (AlexaFluor Goat Anti-Mouse 488 Thermo Fischer Scientific, R37120; AlexaFluor Goat Anti-Rabbit 488 Thermo Fischer Scientific, A32731; AlexaFluor Goat Anti-Rat 488 Abcam, ab150157; AlexaFluor Goat Anti-Mouse 594 Abcam, ab150116; AlexaFluor Goat Anti-Rabbit 594 Thermo Fischer Scientific, A11012; AlexaFluor Goat Anti-Rat 568 Thermo Fischer Scientific, A-11077; AlexaFluor Goat Anti-Rabbit 647 Thermo Fischer Scientific, A-27040) for 2 h at room temperature. Nuclear stain, DAPI was added at a concentration of 1:10,000 90 min post-incubation with the secondary antibody. Afterward, sections were mounted onto microscopic slides using immu-mount, covered with coverslips, and imaged using a Zeiss (AX10) or Zeiss LSM980 microscope.

## Treatment of cortical neurons and forebrain organoids

Between days 10 and 20 during cortical differentiation, the B27 supplement (without vitamin A) was replaced with the B27 supplement without antioxidants (B27 supplement-AO, #10889038, Thermo Fisher Scientific) and B27 supplement without antioxidants + 10 μM all-trans Retinoic Acid (ATRA) or 1 μM ferrostatin-1, respectively. During the forebrain organoid generation, organoids were kept in organoid differentiation media without antioxidants (B27 supplement-AO, #10889038, Thermo Fisher Scientific) and 1 μM ferrostatin-1, respectively, from day 18 onwards.

## Image analysis

High-content image analysis of neurons was performed as previously described[18]. Image analysis of spheroids and BODIPY staining, Columbus software (PerkinElmer) was used. Spheroids were detected as "Image region", and morphology properties were calculated (e.g., roundness). For BODIPY staining, the number of green fluorescent objects per well was counted. For analyzing the intensity of antibody staining in organoids, several images of each section from each experimental condition were processed using ImageJ software. For each section, several fields were subjected to intensity measurement, and the mean average intensity of the staining was calculated in ImageJ.

## C. elegans in vivo model

Nematodes were grown at 20 °C on standard Nematode Growth Media (NGM) plates (Agar 2%, Bacto-Peptone 0,25%, NaCl 0,3%, H2O, Cholesterol 4 mM, CaCl2 4 mM, MgSO$_4$ 4 mM, KH$_2$PO$_4$ 4 mM) seeded with *Escherichia coli* OP50 bacteria grown overnight. To induce ferroptosis, NGM plates were supplemented with 100 mM Ferric Ammonium Citrate (FAC) ((#F5879, Sigma) directly into the media before pouring the plates. To rescue the ferroptosis effect, vitamin A (ATRA) (#10023433, Fischer Scientific) 5 μM final, or vitamin E (Trolox) ((#Cay10011659-1, Biomol) 200 μM final, were added to NGM + FAC 100 mM media (0.05-0.02% DMSO final) before pouring the plates. Plates were maintained in the dark during the whole experiment as both FAC and ATRA are sensitive to light.

HW2668 (xeSi300[Peft-3::luc::mCherry:unc-54 3'UTR] II) adult worms were collected with standard M9 buffer, rinsed twice with M9 buffer and treated with standard hypochlorite treatment to isolate embryos. Embryos were then rinsed 3 times with M9 buffer, and about 35 embryos were seeded per plate directly on top of the bacteria, for all conditions. 72 h after seeding, the hatched worms were imaged using a Leica M165 FC Fluorescent Stereo Microscope. Head to tail length of the worms was measured using mCherry (expressed throughout the body in HW2668 animals) with Fiji/ImageJ[51,52].

## Statistical analysis

All statistical testing for significance was performed in GraphPad Prism Software version 9.4 or R version 4.2.0 with the DESeq2 package. We performed Shapiro-Wilk tests on our experiments. For all experiments that passed the normality test, indicating a normal distribution, we used the ANOVA parametric statistic to determine statistical significance. For those experiments that did not pass the Shapiro-Wilk tests, we used the Kruskal-Wallis non-parametric statistic to determine statistical significance. For the *C. elegans* length experiments, we used the Wilcoxon rank sum test with continuity correction. Statistical details of every experiment are described in the figure legends.

The significance of RNAseq analysis was assessed using the Wald Test, and *p*-values were adjusted for multiple hypothesis testing by the Benjamini-Hochberg procedure.

## Ethics

Working with human stem cells H9 (WA09) was approved by the Robert Koch Institute (RKI) under the number AZ 3.04.02/0168.

## Reporting summary

Further information on research design is available in the Nature Portfolio Reporting Summary linked to this article.

# Data availability

All data are available in the article and its Supplementary Information. Western blot full scan images are shown in Supplementary Fig. 7. The RNAseq data (count data) generated in this study are provided in Supplementary Data 3. In addition, fastq files are deposited at NIH BioProject under the number PRJNA1115822. Source data are provided in this paper.

# Code availability

The source code is available at https://github.com/MendenLab/DEx_StemCells_new.

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

## Acknowledgements

We thank Lena Klepper and Stefanie Brandner for their excellent technical assistance. The authors thank Milou Meeuse and Helge Großhans for gifting the unpublished strain HW2668. M.V. was funded by the Deutsche Forschungsgemeinschaft (RTG2668 – project number 435874434, Sachbeihilfe – project number 496872373, Sachbeihilfe – project number 498956525, and Sachbeihilfe – project number 497803923).

## Author contributions

K.H. and M.V. conceptualized, designed, and generally supervised the study; J.T., I.R. and K.H. performed, analyzed, and interpreted the experiments in HT-1080 cells; V.P.N., G.C., J.Tch., H.M.T. and M.V. performed, analyzed and interpreted the stem cell experiments; C.Z. performed the *C. elegans* in vivo experiments and analyzed and interpreted

the data with support from D.S.C., A.G. and M.P.M. analyzed and interpreted the RNAseq data; B.R.S., L.S., D.S.C., M.P.M., M.V., and K.H. supervised individual experiments; J.T., V.P.N., M.V. and K.H. wrote the original draft; J.T., V.P.N., M.V. and K.H. edited the manuscript; all authors read, commented and approved the manuscript for submission.

## Funding

## Competing interests

B.R.S. is an inventor on patents (granted) and patent applications (pending) involving ferroptosis; co-founded and serves as a consultant to ProJenX, Inc. and Exarta Therapeutics; holds equity in Sonata Therapeutics; serves as a consultant to Weatherwax Biotechnologies Corporation and Akin Gump Strauss Hauer & Feld LLP; and receives sponsored research support from Sumitomo Dainippon Pharma Oncology. L.S. is a scientific founder and paid consultant of BlueRock Therapeutics, a scientific founder of DaCapo Brain Science, and an inventor on patents (granted) owned by MSKCC on the differentiation of human pluripotent stem cells into specific neurons. J.T. and K.H. are inventors on a patent application (pending) involving ferroptosis. The remaining authors declare no competing interests.
