## [Peer Review File · Nature Communications]

Suppression of ferroptosis by vitamin A or radical-trapping antioxidants is essential for neuronal developmentREVIEWER COMMENTS

Reviewer #1 (Remarks to the Author):

Tschuck, Padmanabhan, and their colleagues propose that the suppression of ferroptosis is a critical factor in promoting successful neuronal development. They report that the removal of antioxidants hampers neuronal development and disrupts the laminar organization in cortical organoids. However, this impairment can be effectively reversed either by inhibiting ferroptosis with ferrostatin-1 or by supplementing with vitamin A, specifically its active metabolite, all-trans retinoic acid (ATRA). Notably, the "anti-ferroptotic" effects of ATRA do not appear to result from direct antioxidant properties. Rather, ATRA activates a nuclear receptor complex, leading to an increase in the expression of ferroptosis inhibitors.

The authors' discovery unveils an unexpected role of vitamin A in coordinating the expression of essential cellular regulators of ferroptosis. This underscores the significance of ferroptosis suppression in facilitating the proper maturation of neurons and the establishment of laminar organization in cortical organoids.

While this study presents intriguing findings with broad implications (it seems well executed, analyzed and the major conclusions are supported by the data – but see my comment regarding n-numbers and statistics), it's essential to acknowledge the following limitations in the current version of the manuscript.

(1) Recent research has uncovered a connection between Retinoid Metabolism and ferroptosis, which does bring into question the novelty of this aspect of the study. While the examination of direct antioxidant properties is certainly valid (and direct experimental evidence in the context of brain development is missing), further investigations are needed to directly assess the significance of ATRA-induced alterations in GPX4, SDP1, DCH1, as well as SCSL4, SCDF1, and PPARα in neural differentiation and cortical lamination.

(2) Along this line, a more thorough evaluation of the influence of ATRA on neuronal differentiation and cortical lamination, especially within the context of ferroptosis, is required. The second part of the study exclusively employs ferrostatin-1 as a single intervention to validate a major conclusion of the study. What about ATRA?

(3) Which cell types are affected by ferroptosis during neurogenesis? RNA sequencing approaches (single nuclei or spatial transcriptomics) may help deciphering and better understanding the impact of ATRA on ferroptosis and development.

(4) The applicability of these findings to in vivo models remains unexplored, raising questions about whether the outcomes are specific to the culture conditions, cell lines/models investigated in this study.

(5) Concerns arise regarding the limited sample size of $n=3$ in many experiments and the use of parametric statistics (for multiple comparisons). How did the authors determine proper n -numbers prior to executing these experiments? Power analysis? Please always provide details, including in the figure legends, regarding what “ n ” specifically refers to, the number of independent experiments conducted, and how the authors verified the normal distribution of the data (justifying parametric statistics). Why present mean \pm SD with three data points?

(6) Regarding lines 95-97: I'm a bit unclear about this point. Would we anticipate similar or more intricate and branched axons when there are fewer cells present? Additionally, I'm curious about the suitability of MAP2 staining for evaluating axon morphologies. Does a reduction in MAP2 necessarily equate to neuronal cell loss?

Reviewer #2 (Remarks to the Author):

In this Ms, the authors aimed to investigate whether the exclusion of the antioxidants would affect neurogenesis and if vitamin A could compensate for these effects caused by antioxidants deficiency. They found that vitamin A restores neuronal differentiation in the condition without antioxidants. Mechanistically, vitamin A inhibited ferroptosis by transcriptionally upregulated anti-ferroptotic regulators, such as FSP1 and ACSL3, through its receptor RAR/RXR. Ferroptosis inhibition indeed promoted neuronal differentiation by adding Fer-1 into the culture medium of lack of antioxidants when differentiated H9 cells into cortical neurons. They also observed impaired laminar organization of cortical organoids without antioxidants, which was reversed by inhibition of ferroptosis. While the study emphasized the importance of antioxidants during early neurogenesis and established the protected role of vitamin A in ferroptosis, a number of concerns need to be elucidated.

-As the recent study “Retinol Saturase Mediates Retinoid Metabolism to Impair a Ferroptosis Defense System in Cancer Cells” has elucidated the mechanism of vitamin A to protect cells from ferroptosis, the work here may be less significant to the field of ferroptosis.

-The aforementioned study demonstrated vitamin A could act as the radical-trapping antioxidant. However, the authors here did not observe such phenomenon, which leads to contradictory results that need to be further discussed.

-Result 1: The authors showed an increase in RARB and a decrease in lipid peroxidation after adding vitamin A. Furthermore, they showed that MAP2 was decreased in -AO+vAlo treated cells while observing similar levels of axon length, branching, and extremities across all three conditions. Ultimately, the authors concluded that "cell death under this condition due to lack of antioxidants rather than impaired neuronal differentiation". However, it should be noted that the results presented in this section do not strongly support their conclusion. Therefore, further analysis regarding the viability and proliferative capacity of neurons under these three different conditions is necessary.

-The authors confirmed the increased expression of RARB and BCO2 in vitamin A dose-dependent manner. However, the higher lipid peroxidation was only reversed in the condition of -AO+vAhi, not in -AO+vAlo. Given that the study ultimately concluded that vitamin A inhibits ferroptosis by reducing lipid peroxidation via its receptor RAR, why low concentration of vitamin A could not reduce lipid peroxidation? Furthermore, why wasn't the condition without antioxidant and vitamin A (-AO) utilized?

-Result 2: As vitamin A upregulates anti-ferroptotic regulators significantly in cortical neurons using qRT-PCR, why these anti-ferroptotic regulators had not been found in the RNA-Seq data? Additionally, it remains unclear why RARB and BCO2 levels increase upon vitamin A supplementation.

-Result 3: By using qRT-PCR in HT1080 and immature neuron, the authors confirmed the transcriptional increase of anti-ferroptotic regulators under vitamin A treatment. The protein level of these anti-ferroptotic regulators by western blotting is needed. And whether the treatment of vitamin A impact the mRNA or protein level of other pro-ferroptotic regulators such as ACSL4 and LOXs ?

Reviewer #3 (Remarks to the Author):

Tschuck et al discovered a ferroptosis suppressing role for vitamin A during neuronal development, which acts through retinoid receptors.

Study looks overall well performed and reads well. However, there is novelty issue.

1. Indirect anti-oxidant role of vit A is known, as also acknowledge in discussion by the authors.
2. Lack of any in vivo data illustrating importance of vit A in neuronal development.

Some considerations to improve manuscript:

- “the anti-ferroptotic impact of ATRA was similar to ferrostatin-1 (Fer-1), a gold standard ferroptosis inhibitor5 (Extended Data Fig. 2b,c).” This is overstatement, protection is less than Fer-1 while using 10x higher dose. The lower potency is also illustrated in Fig 2c.
- For FENIX (Fig 1f) and DDPH assay (Extended Fig 2e), data are typically represented as curves which visualize consumption of substrate, not bar graphs.
- How do the authors explain difference observed in Fig 2b and c. C11-bodipy oxidation is completely blocked by ARTRA, but not formation of 4HNE adducts.
- There seems to be difference in reproducibility of neuronal differentiation. Compare Fig 1d with Fig 3c. Number of objects vs segments is completely different. Please explain.
- Although TfR1 is upregulated during ferroptosis, one cannot use upregulation of TfR1 as indication for ferroptosis (Fig 3f). Please use more specific techniques 4HNE staining or oxlipidomics.

REVIEWER COMMENTS

Reviewer #1 (Remarks to the Author):

Tschuck, Padmanabhan, and their colleagues propose that the suppression of ferroptosis is a critical factor in promoting successful neuronal development. They report that the removal of antioxidants hampers neuronal development and disrupts the laminar organization in cortical organoids. However, this impairment can be effectively reversed either by inhibiting ferroptosis with ferrostatin-1 or by supplementing with vitamin A, specifically its active metabolite, all-trans retinoic acid (ATRA). Notably, the "anti-ferroptotic" effects of ATRA do not appear to result from direct antioxidant properties. Rather, ATRA activates a nuclear receptor complex, leading to an increase in the expression of ferroptosis inhibitors.

The authors' discovery unveils an unexpected role of vitamin A in coordinating the expression of essential cellular regulators of ferroptosis. This underscores the significance of ferroptosis suppression in facilitating the proper maturation of neurons and the establishment of laminar organization in cortical organoids.

While this study presents intriguing findings with broad implications (it seems well executed, analyzed and the major conclusions are supported by the data – but see my comment regarding n-numbers and statistics), it's essential to acknowledge the following limitations in the current version of the manuscript.

We thank the reviewer for this very positive assessment of our study.

(1) Recent research has uncovered a connection between Retinoid Metabolism and ferroptosis, which does bring into question the novelty of this aspect of the study. While the examination of direct antioxidant properties is certainly valid (and direct experimental evidence in the context of brain development is missing), further investigations are needed to directly assess the significance of ATRA-induced alterations in GPX4, SDP1, DCH1, as well as SCSL4, SCDF1, and PPARa in neural differentiation and cortical lamination.

The manuscript "Retinol Saturase Mediates Retinoid Metabolism to Impair a Ferroptosis Defense System in Cancer Cells" published in Cancer Research was published 3 months after we had posted our work on BioRxiv (<https://www.biorxiv.org/content/10.1101/2023.04.05.535746v1>). Thus, our study and theirs evolved in parallel, underscoring the credibility and importance of the findings.

Mechanistically, the Cancer Research paper suggests that retinal, retinol and ATRA have radical-trapping antioxidant (RTA) activity. We observed that the mechanism is more complex, as shown in new Figure 4e-h and 5f). We now demonstrate that retinal and retinol do have RTA activity, but ATRA does not act as an antioxidant (new Figure 4e-h). Rather, ATRA binds the RAR and activates the expression of key anti-ferroptotic regulators, i.e., GPX4, FSP1, GCH1, ACSL3, SCD1 and PPARa (Figure 5e and new Figure 5f). These genes either remove lipid hydroperoxides (GPX4), generate endogenous antioxidants to scavenge lipid radicals (FSP1 and GCH1), or induce lipid remodeling toward MUFA-rich membranes (ACSL3, SCD1 and PPARa). These three pillars of anti-ferroptotic defense reduce lipid peroxidation and thus inhibit ferroptotic cell death (1, 2). Therefore, under -AO conditions, cells accumulate lipid peroxidation (Figure 1c, 2b, 2g, Supplementary Figure 2a, 2d) leading to cell death. ATRA supplementation transcriptionally upregulates the expression of the ferroptosis inhibitory regulators (Figure 5f) to facilitate neuronal differentiation. We added these new

findings to the manuscript and extended the discussion on this aspect describing the dual mode of action of vitamin A derivatives.

Regarding „examination of direct antioxidant properties is certainly valid (and direct experimental evidence in the context of brain development is missing)”: in this study we show that cortical development and forebrain organoid formation are impaired in -AO conditions (**Figure 1-3**). Physiologically, a recent study showed that vitamin E is necessary for zebrafish nervous system development (3). Also, vitamin E deficiency leads to accumulation of lipid peroxides in the zebrafish nervous system and impairs cognitive functions (4), which supports our findings.

(2) Along this line, a more thorough evaluation of the influence of ATRA on neuronal differentiation and cortical lamination, especially within the context of ferroptosis, is required. The second part of the study exclusively employs ferrostatin-1 as a single intervention to validate a major conclusion of the study. What about ATRA?

It is well described that vitamin A plays a critical role in various aspects of neuronal development by regulating gene expression, cell differentiation, neurite outgrowth, synaptogenesis, and neurotransmitter synthesis via its active metabolite ATRA, which has been described in neuronal cell models as well as in *in vivo* studies (5-7). Studies have also shown that ATRA promotes the proliferation of adult hippocampal neural stem and progenitor cells (8). ATRA has also been described as a crucial factor in the early stages of neurogenesis and survival *in vivo* (9).

Therefore, we have now restructured the manuscript and consolidated the neuro part, as we think this is easier for the reader to understand the major novelties:

1. During early neurogenesis ferroptosis has to be prevented in order to generate mature neurons.
2. ATRA can inhibit ferroptosis and thereby enhance mature neuron development. However, this can also be achieved by antioxidants such as vitamin E or ferrostatin-1.

Thus, while in Figure 1 we show that vitamin A can help overcome lipid peroxidation to generate neurons, the studies with ferrostatin-1 as the gold standard to inhibit ferroptosis (now Figure 2 and 3) is essential to demonstrate the occurrence of ferroptosis. Ferrostatin-1 also inhibits the lipid peroxidation (now shown with C11-BODIPY and 4-HNE staining), and restores MAP2 positive neurons, further proving that ferroptotic events had been inhibited.

Regarding the organoid data, we have done many experiments to ATRA to overcome ferroptosis and then investigate laminar patterning. Unfortunately, we could not generate forebrain organoids upon ATRA supplementation as they fell apart. ATRA not only inhibits ferroptosis, but also orchestrates a battery of developmental genes necessary of the organoid formation (5-7). Hence, we could not determine the narrow window of ferroptosis protection, but not mis-regulating developmental genes. Nevertheless, the data with ferrostatin-1 demonstrates that inhibition of ferroptosis is important for organoid organization, which is a key take home message of our study.

(3) Which cell types are affected by ferroptosis during neurogenesis? RNA sequencing approaches (single nuclei or spatial transcriptomics) may help deciphering and better understanding the impact of ATRA on ferroptosis and development.

We thank the reviewer for this suggestion, which we agree would be very interesting. However, we believe it is beyond the scope of this study. In fact, we are in the process of obtaining funding for these very cost-intense experiments to be able to perform them as a follow-up study in the future. This aspect was is mentioned in the discussion as a future perspective: "Future research will decipher the exact cell types affected by ferroptosis during neurogenesis by applying single cell and spatial transcriptomics."

(4) The applicability of these findings to *in vivo* models remains unexplored, raising questions about whether the outcomes are specific to the culture conditions, cell lines/models investigated in this study.

We agree with this reviewer that it is interesting to follow up our conclusions in relevant animal models. However, our study **aimed at underpinning the mechanism** by which antioxidants (*e.g.*, vitamin E or ferrostatin-1) or vitamin A counteract ferroptosis to ensure proper neuronal development. We were able to provide strong and convincing evidence that activation of RAR by ATRA orchestrates the transcriptional upregulation of ferroptosis gatekeepers to prevent lipid peroxidation. Moreover, Retinal and Retinol act as RTAs to prevent ferroptosis (new data!).

The fact that ATRA facilitates neurogenesis (5-7), and that antioxidants are needed for neurodevelopment are known and we have discussed these points in the discussion. However, the mechanism of why this is the case is new and the core of this study. We have used a 2D stem cell differentiation, a forebrain organoid model, HT-1080 cells as one of the best studied ferroptosis-related cell line and HT-1080-derived spheroids to uncover that ferroptosis needs to be suppressed to guarantee proper neurodevelopment. Here, ferroptosis can be suppressed by antioxidants (Fer-1) or vitamin A to generate mature neurons. *In vivo*, a recent study showed that vitamin E is necessary for zebrafish nervous system development (3). Also, vitamin E deficiency leads to accumulation of lipid peroxides in the zebrafish nervous system and impairs cognitive functions (4). This supports our finding that early neuronal development has to suppress ferroptosis. As our institution is committed to the 3R principles (replace, reduce, refine) regarding animal testing, we would like to refrain from *in vivo* studies that would not add a lot of insight to the mechanistic understanding of this study.

In order to provide further *in vivo* evidence of our findings, we added *in vivo* data in *C. elegans* showing that vitamin A (ATRA) indeed ameliorates developmental growth induced by iron-overload and subsequent ferroptosis (new Figure 4a). This new data further strengthens the *in vivo* relevance of our study.

(5) Concerns arise regarding the limited sample size of $n=3$ in many experiments and the use of parametric statistics (for multiple comparisons). How did the authors determine proper n -numbers prior to executing these experiments? Poweranalysis? Please always provide details, including in the figure legends, regarding what “ n ” specifically refers to, the number of independent experiments conducted, and how the authors verified the normal distribution of the data (justifying parametric statistics). Why present mean \pm SD with three data points?

We agree that the figure legends in the original submission were incomplete and improved them in the revised version. We now provide information about the number of replicates and the exact statistical test. We also provide the exact p values in the figures. In our experiments we always performed at least 3 biologically independent experiments, in some cases also more. These were done independently on different days, thus being true biological replicates. In fact, this is a very standard procedure and leading studies in the field use $n = 3$ or 4 biologically independent experiments to show biological effects (10-14). In several cases, the biological replicates in our study are composed of several technical replicates as well.

We are showing mean \pm SD, but in the revised version also included all individual data points so that the reader can see the exact distribution of the individual data points.

We performed Shapiro-Wilk tests on our experiments. The Shapiro-Wilk test provides a strong test of normality that can be used with sample sizes as small as 3 data points. It can be used to check whether the data violate the assumption of normal distribution that is required for parametric tests. For experiments that passed the normality test indicating a normal distribution, we used the ANOVA parametric statistic to determine statistical significance. For those experiments that did not pass the Shapiro-Wilk tests, we used the Kruskal-Wallis non-parametric statistic to determine statistical significance. This has been also added to the Method section.

(6) Regarding lines 95-97: I'm a bit unclear about this point. Would we anticipate similar or more intricate and branched axons when there are fewer cells present? Additionally, I'm curious about the suitability of MAP2 staining for evaluating axon morphologies. Does a reduction in MAP2 necessarily equate to neuronal cell loss?

We apologize for not being clear enough. We failed to label “mean per well”. So, this is not the cumulative absolute number of neurite lengths and branches. It reflects the average, which obviously does not take into account the number of cells. We did this on purpose, because obviously (as this reviewer correctly points out) the -AO samples would always have lower counts because there are fewer cells due to ferroptotic cell death. We wanted to show that for those cells, which managed to generate mature neurons under low antioxidant conditions, possibly due to intrinsic upregulation of ferroptosis gatekeepers, the neurons have a “normal” phenotype with similar length and branching. One of our future projects is to elucidate the mechanism by which these cells survive ferroptosis despite being under -AO conditions.

MAP2 is a neuron-specific cytoskeletal protein and a suitable marker for neuronal cells to study morphological (cytoskeletal) changes. We agree that it is more suitable for visualizing dendrites, but overall, it stains the larger morphology of neurons. The neurite length and the branches measured in this work are mostly the dendrites, but provide enough information to determine neuronal morphology. We have corrected this in the manuscript by changing axon length to neurite length. The reduction of MAP2-positive cells (not in MAP2 fluorescence intensity) together with the reduced number of healthy nuclei is equivalent to neuronal cell loss. We also observe no changes in Ki67 staining (**new Supplementary Fig. 2b**) as a marker of proliferation.

In the revised manuscript we have labeled the y-axes as “mean per well” (see **Supplementary Fig. 2c**) and also clarified the main text.

Reviewer #2 (Remarks to the Author):

In this Ms, the authors aimed to investigate whether the exclusion of the antioxidants would affect neurogenesis and if vitamin A could compensate for these effects caused by antioxidants deficiency. They found that vitamin A restores neuronal differentiation in the condition without antioxidants. Mechanistically, vitamin A inhibited ferroptosis by transcriptionally upregulated anti-ferroptotic regulators, such as FSP1 and ACSL3, through its receptor RAR/RXR. Ferroptosis inhibition indeed promoted neuronal differentiation by adding Fer-1 into the culture medium of lack of antioxidants when differentiated H9 cells into cortical neurons. They also observed impaired laminar organization of cortical organoids without antioxidants, which was reversed by inhibition of ferroptosis. While the study emphasized the importance of antioxidants during early neurogenesis and established the protected role of vitamin A in ferroptosis, a number of concerns need to be elucidated.

-As the recent study “Retinol Saturase Mediates Retinoid Metabolism to Impair a Ferroptosis Defense System in Cancer Cells” has elucidated the mechanism of vitamin A to protect cells from ferroptosis, the work here may be less significant to the field of ferroptosis.

The manuscript “Retinol Saturase Mediates Retinoid Metabolism to Impair a Ferroptosis Defense System in Cancer Cells” published in Cancer Research was published 3 months after we had already posted our work on BioRxiv. Thus, our study and theirs evolved in parallel, underscoring the credibility and importance of the findings.

Mechanistically, the Cancer Research paper suggests that retinal, retinol and ATRA have radical-trapping antioxidant (RTA) activity, which we don't see. The mechanism is more complex as we demonstrate, because retinal and retinol do indeed have RTA activity, but ATRA does not act as an antioxidant (**new Figure 4e-h**). Rather, ATRA binds the RAR and activates the expression of key anti-ferroptotic regulators, *i.e.*, GPX4, FSP1, GCH1, ACSL3, SCD1 and PPARa (**Figure 5e and new Figure 5f**). These genes either remove lipid hydroperoxides (GPX4), generate endogenous antioxidants to scavenge lipid radicals (FSP1 and GCH1), or induce lipid remodeling toward MUFA-rich membranes (ACSL3, SCD1 and PPARa). These three pillars of anti-ferroptotic defense reduce lipid peroxidation and thus inhibit ferroptotic cell death (1, 2). Therefore, under -AO conditions, cells accumulate lipid peroxidation (**Figure 1c, 2b, 2g, Supplementary Figure 2a, 2d**) leading to cell death. ATRA supplementation transcriptionally upregulates the expression of the ferroptosis inhibitory regulators (**Figure 5f**) to facilitate neuronal differentiation. We now extended the discussion on this aspect describing the dual mode of action of vitamin A derivatives.

-The aforementioned study demonstrated vitamin A could act as the radical-trapping antioxidant. However, the authors here did not observe such phenomenon, which leads to contradictory results that need to be further discussed.

As indicated before, we now demonstrate that retinal and retinol have radical trapping antioxidant (RTA) activity, while ATRA does not act as an RTA (**new Figure 4f**), but rather binds the RAR and activates the expression of key anti-ferroptotic regulators, *i.e.*, GPX4, FSP1, GCH1, ACSL3, SCD1 and PPARa (**new Figure 5d, 5e**). By using the pan-RAR inhibitor AGN 193109 (RARI) we could demonstrate that the ATRA-mediated anti-ferroptotic effect is indeed receptor-dependent and not mediated by radical trapping. Hence, we now provide a complete picture of the dual mode of action of vitamin A derivatives.

-Result 1: The authors showed an increase in RARB and a decrease in lipid peroxidation after adding vitamin A. Furthermore, they showed that MAP2 was decreased in -AO+vAlo treated cells while observing similar levels of axon length, branching, and extremities across all three conditions. Ultimately, the authors concluded that "cell death under this condition due to lack of antioxidants rather than impaired neuronal differentiation". However, it should be noted that the results presented in this section do not strongly support their conclusion. Therefore, further analysis regarding the viability and proliferative capacity of neurons under these three different conditions is necessary.

In our study we show that the number of objects is reduced based on nuclear stain. Obviously, this could be cell death or proliferation. However, as ferrostatin-1 largely restores the number of healthy nuclei (201 with AO, 63 minus AO, 159 minus AO plus Fer-1) this argues for ferroptotic cell death rather than proliferative defects. Moreover, we now tested Ki67 levels at day-20 and these levels are very similar (**new Supplementary Figure 2b**), again arguing against a proliferative effect.

-The authors confirmed the increased expression of RARB and BCO2 in vitamin A dose-dependent manner. However, the higher lipid peroxidation was only reversed in the condition of -AO+vAhi, not in -AO+vAlo. Given that the study ultimately concluded that vitamin A inhibits ferroptosis by reducing lipid peroxidation via its receptor RAR, why low concentration of vitamin A could not reduce lipid peroxidation? Furthermore, why wasn't the condition without antioxidant and vitamin A (-AO) utilized?

As we show in Figure 5f and 5g, the -AO+vA^{lo} is not enough to upregulate the ferroptosis inhibitory regulators GPX4, FSP1, GCH1, ACSL3. Only at higher levels of ATRA (-AO+vA^{hi}) these genes are elevated. The elevation of these genes is necessary to revert lipid hydroperoxides and to scavenge lipid radicals. Unfortunately, we cannot specify what the quantity of “low” is, as the supplier Thermo does not disclose the exact amount. Thus, we can only say that “Therefore, our study reveals that above a certain threshold, vitamin A increases expression of essential cellular gatekeepers of lipid peroxidation.” This observed threshold might stem from the mode of action of RAR: in the non-activated state, it remains DNA-bound and complexed by transcription corepressors. Upon ligand binding, the repressors dissociate and instead, co-activators are bound. Since RAR activation regulates a variety of tightly controlled target genes, it seems plausible that basal levels of ATRA are not enough to significantly upregulate anti-ferroptotic genes (15).

Also, we couldn't generate B27 supplements without vitamin A, as this is not provided by the supplier, and the exact amount of the ingredients is not specified in order for us to mix our own -AO without vitamin A B27. However, we feel that the vitamin A low vs high conditions have also their advantage in suggesting dosage effects in regulating ferroptosis inhibitory regulators.

-Result 2: As vitamin A upregulates anti-ferroptotic regulators significantly in cortical neurons using qRT-PCR, why these anti-ferroptotic regulators had not been found in the RNA-Seq data? Additionally, it remains unclear why RARB and BCO2 levels increase upon vitamin A supplementation.

The qRT-PCR was performed with +AO and -AO+vA^{lo}. The -AO+vA^{lo} condition was sufficient to upregulate RARB and BCO2 as direct target genes of the Retinoic Acid Receptor, but not sufficient enough to upregulate the anti-ferroptotic genes. Fig. 5f shows that only -AO+vA^{hi} induces a robust upregulation of the anti-ferroptotic regulators. This is consistent with our findings that the -AO+vA^{lo} is not sufficient to reduce lipid peroxidation (Fig. 1c) and is not sufficient to overcome ferroptosis to generate mature neurons (Fig. 1d). Nevertheless, we have now re-checked our RNAseq data and can see that even with -AO+vA^{hi} there is a slight increase in GPX4 and FSP1 mRNA levels. These data are now shown in the **new Supplementary Fig 1b**.

-Result 3: By using qRT-PCR in HT1080 and immature neuron, the authors confirmed the transcriptional increase of anti-ferroptotic regulators under vitamin A treatment. The protein level of these anti-ferroptotic regulators by western blotting is needed. And whether the treatment of vitamin A impact the mRNA or protein level of other pro-ferroptotic regulators such as ACSL4 and LOXs ?

We agree that this is important. In the revised manuscript, we now provide Western Blot data showing the upregulation of GPX4, FSP1, ACSL3, and SCD1 in HT-1080 cells upon ATRA treatment (**new Figure 5e**), as well as GPX4, FSP1 and SCD1 in -AO+vA^{hi} immature neurons versus -AO+vA^{lo} (**new Figure 5g**). We also investigated the protein levels of ACSL4 and LPCAT3 (**new Fig. 5e**), two of the most important enzymes for activation and incorporation of PUFA-PLs into the membrane. Interestingly, levels of ACSL4 and LPCAT3 were reduced in ATRA-treated HT-1080 cells. This is consistent with the concept of ferroptosis inhibition by ATRA, but the mechanism by which ATRA reduces ACSL4 and LPCAT3 levels is a matter of future research.

Reviewer #3 (Remarks to the Author):

Tschuck et al discovered a ferroptosis suppressing role for vitamin A during neuronal development, which acts through retinoid receptors.

Study looks overall well performed and reads well. However, there is novelty issue.
1. Indirect anti-oxidant role of vit A is known, as also acknowledge in discussion by the authors.

The manuscript "Retinol Saturase Mediates Retinoid Metabolism to Impair a Ferroptosis Defense System in Cancer Cells" published in Cancer Research was published 3 months after we had already posted our work on BioRxiv. Thus, our study and theirs evolved in parallel, underscoring the credibility of the findings.

Mechanistically, the Cancer Research paper suggests that retinal, retinol and ATRA have radical-trapping antioxidant (RTA) activity, which we don't see. The mechanism that we demonstrate is more complex, because retinal and retinol do indeed have RTA activity, but ATRA does not act as an antioxidant (**new Figure 4e-h**). Rather, ATRA rather binds the RAR and activates the expression of key anti-ferroptotic regulators, *i.e.*, GPX4, FSP1, GCH1, ACSL3, SCD1 and PPAR α (**Figure 5e and new Figure 5f**). These genes either remove lipid hydroperoxides (GPX4), generate endogenous antioxidants to scavenge lipid radicals (FSP1 and GCH1), or induce lipid remodeling toward MUFA-rich membranes (ACSL3, SCD1 and PPAR α). These three pillars of anti-ferroptotic defense reduce lipid peroxidation and thus inhibit ferroptotic cell death (1, 2). Therefore, under -AO conditions, cells accumulate lipid peroxidation (**Figure 1c, 2b, 2g, Supplementary Figure 2a, 2d**) leading to cell

death. ATRA supplementation transcriptionally upregulates the expression of the ferroptosis inhibitory regulators (**Figure 5f**) to facilitate neuronal differentiation. We now extended the discussion on this aspect describing the dual mode of action of vitamin A derivatives.

2. Lack of any *in vivo* data illustrating importance of vit A in neuronal development.

We agree with the reviewers that it is interesting to follow up our conclusions in relevant animal models. However, our study **aimed at underpinning the mechanism of ferroptosis inhibition for proper neuronal development by antioxidants** (e.g., vitamin E or ferrostatin-1) **or vitamin A**. We were able to provide strong and convincing evidence that activation of RAR by ATRA orchestrates the transcriptional upregulation of ferroptosis gatekeepers to prevent lipid peroxidation. Moreover, retinal and retinol act as RTAs to prevent ferroptosis. We performed 2D stem cell differentiations as well as human brain organoid studies to answer this question in human relevant systems. Human brain organoids have become increasingly important in research in recent years, because they closely mimic the development and function of the human brain (brain-in-a-dish). For these reasons, they are often a good replacement for *in vivo* mouse studies.

In fact, getting approval in Germany to do *in vivo* mouse studies where many information on the *in vivo* relevance (RA in neurodevelopment) is known (5-7), and where organoid models are available, which we employed, is very difficult. As our institution is committed to the 3R principles (replace, reduce, refine) regarding animal testing, we would like to refrain from *in vivo* studies that would not add a lot insight to the mechanistic understanding of this study.

In order to provide further *in vivo* evidence of our findings, we added *in vivo* data in *C. elegans* showing that vitamin A (ATRA) indeed ameliorates developmental growth induced by iron-overload and subsequent ferroptosis (new Figure **4a**). This new data further strengthens the *in vivo* relevance of our study.

Some considerations to improve manuscript:

- “the anti-ferroptotic impact of ATRA was similar to ferrostatin-1 (Fer-1), a gold standard ferroptosis inhibitor5 (Extended Data Fig. 2b,c).” This is overstatement, protection is less than Fer-1 while using 10x higher dose. The lower potency is also illustrated in Fig 2c.

We have now clarified that the similar effect is at different concentrations.

- For FENIX (Fig 1f) and DPPH assay (Extended Fig 2e), data are typically represented as curves which visualize consumption of substrate, not bar graphs.

In this study, we have not used the FENIX assay. The two experiments are cell-free BODIPY and DPPH according to our previous publications in ACS Central Science (2020) and Nature Communications (2023) (13, 16). These assays reliably detect whether a compound has radical-trapping antioxidant capacity.

- How do the authors explain difference observed in Fig 2b and c. C11-bodipy oxidation is completely blocked by ARTRA, but not formation of 4HNE adducts.

By looking at the quantification, the level of inhibition of lipid peroxidation is similar between C11-BODIPY and 4-HNE. We illustrate only one of the replicates and show the quantification of 3 biologically independent experiments, which ultimately look similar.

- There seems to be difference in reproducibility of neuronal differentiation. Compare Fig 1d with Fig 3c. Number of objects vs segments is completely different. Please explain.

Number of objects is mainly determined by the number of neurons with nuclear stain (DAPI). Segments represent regions of the neuronal outreach (with MAP2 staining mostly dendrites). In **Figure 1d** we show the number of objects. In **Figure (now) 2c**, we have the number of healthy nuclei in the mini table – also showing a reduction in -AO conditions – and the segments as additional information of lack of proper neuronal development in the graph. Therefore, number of objects and segments are not the same. We clarified this in the manuscript.

- Although TfR1 is upregulated during ferroptosis, one cannot use upregulation of TfR1 as indication for ferroptosis (Fig 3f). Please use more specific techniques 4HNE staining or oxlipidomics.

We have previously published that TfR1 is a valid marker for ferroptosis (17), which is now widely accepted and used by many researchers (> 400 citations). However, in order to further strengthen our interpretation, we performed a series of additional experiments using 4-HNE staining:

First, we tested 4-HNE levels in the neurons from 2D cultures at day-20. Similar to C11-BODIPY, levels of 4-HNE are increased in -AO conditions and reduced again by using Fer-1 or vit A (**new Supplementary Figure 2a**).

We also stained sections of day-60 organoids with an anti 4-HNE antibody. Again, levels of 4-HNE are increased in -AO conditions and reduced by using Fer-1 (**new Figure 2g**).

Finally, we dissociated day-40 organoids (n=3 independent organoids) and stained the single cells with a 4-HNE antibody and analyzed the cells using flow cytometry. An increase in 4-HNE levels was detected in -AO condition, which was reverted to background levels upon Fer-1 treatment (**new Supplementary Figure 2d**).

References Rebuttal letter:

1. K. Hadian, B. R. Stockwell, SnapShot: Ferroptosis. *Cell* **181**, 1188-1188.e1181 (2020).
2. B. R. Stockwell, Ferroptosis turns 10: Emerging mechanisms, physiological functions, and therapeutic applications. *Cell* **185**, 2401-2421 (2022).
3. B. Head, J. La Du, R. L. Tanguay, C. Kioussi, M. G. Traber, Vitamin E is necessary for zebrafish nervous system development. *Sci Rep* **10**, 15028 (2020).
4. M. McDougall, J. Choi, L. Truong, R. Tanguay, M. G. Traber, Vitamin E deficiency during embryogenesis in zebrafish causes lasting metabolic and cognitive impairments despite refeeding adequate diets. *Free Radic Biol Med* **110**, 250-260 (2017).
5. M. A. Lane, S. J. Bailey, Role of retinoid signalling in the adult brain. *Prog Neurobiol* **75**, 275-293 (2005).
6. M. Maden, Retinoic acid in the development, regeneration and maintenance of the nervous system. *Nat Rev Neurosci* **8**, 755-765 (2007).
7. A. Janesick, S. C. Wu, B. Blumberg, Retinoic acid signaling and neuronal differentiation. *Cell Mol Life Sci* **72**, 1559-1576 (2015).
8. S. Mishra, K. K. Kelly, N. L. Rumian, J. A. Siegenthaler, Retinoic Acid Is Required for Neural Stem and Progenitor Cell Proliferation in the Adult Hippocampus. *Stem Cell Reports* **10**, 1705-1720 (2018).
9. S. Jacobs *et al.*, Retinoic acid is required early during adult neurogenesis in the dentate gyrus. *Proc Natl Acad Sci U S A* **103**, 3902-3907 (2006).
10. F. P. Freitas *et al.*, 7-Dehydrocholesterol is an endogenous suppressor of ferroptosis. *Nature* **626**, 401-410 (2024).
11. Z. Li *et al.*, Ribosome stalling during selenoprotein translation exposes a ferroptosis vulnerability. *Nature chemical biology* **18**, 751-761 (2022).
12. E. Mishima *et al.*, A non-canonical vitamin K cycle is a potent ferroptosis suppressor. *Nature* **608**, 778-783 (2022).
13. J. Tschuck *et al.*, Farnesoid X receptor activation by bile acids suppresses lipid peroxidation and ferroptosis. *Nat Commun* **14**, 6908 (2023).
14. P. S. Upadhyayula *et al.*, Dietary restriction of cysteine and methionine sensitizes gliomas to ferroptosis and induces alterations in energetic metabolism. *Nat Commun* **14**, 1187 (2023).
15. C. M. Klinge, D. L. Bodenner, D. Desai, R. M. Niles, A. M. Traish, Binding of type II nuclear receptors and estrogen receptor to full and half-site estrogen response elements in vitro. *Nucleic Acids Res* **25**, 1903-1912 (1997).
16. V. A. N. Kraft *et al.*, GTP Cyclohydrolase 1/Tetrahydrobiopterin Counteract Ferroptosis through Lipid Remodeling. *ACS central science* **6**, 41-53 (2020).
17. H. Feng *et al.*, Transferrin Receptor Is a Specific Ferroptosis Marker. *Cell Rep* **30**, 3411-3423.e3417 (2020).

REVIEWERS' COMMENTS

Reviewer #1 (Remarks to the Author):

The authors have addressed all my comments effectively. They provided new data and clarifications, enhancing the manuscript's overall quality. Their responses adequately covered concerns about the novelty of their findings, the role of ATRA in neuronal differentiation, the mechanisms of ferroptosis suppression, and the in vivo relevance of their results. My question regarding n-numbers has been also addressed.

- The inclusion of *C. elegans* data supports the in vivo relevance of their findings; further justification of the relevance of this model to human neurodevelopment might strengthen the manuscript. Yet, the authors have addressed potential limitations and outlined future research directions in their discussion.

- A final proofread to catch any remaining typographical errors would be helpful. For example, "...neuronal development has to suppress ferroptosis by antioxidants" should be "...neuronal development requires the suppression of ferroptosis by antioxidants." Also, ensure all figure legends are detailed and clear.

Overall, the authors have successfully addressed all comments and provided additional data to support their claims. The study offers valuable insights into the role of ferroptosis suppression in neuronal development and the mechanisms of vitamin A in this process.

I recommend the revised manuscript for publication after minor typographical corrections and a final proofreading.

Reviewer #2 (Remarks to the Author):

The authors have addressed all my concerns.

Reviewer #3 (Remarks to the Author):

This revised manuscript is strengthened and improved related to mode-of-action of retinal, retinol and ATRA, however in vivo relevance (impact) of this mechanism is still not convincingly shown.

Experiment in *C.elegans* only shows minor effect, questioning its importance as major anti-ferroptosis mechanism.

REVIEWERS' COMMENTS

Reviewer #1 (Remarks to the Author):

The authors have addressed all my comments effectively. They provided new data and clarifications, enhancing the manuscript's overall quality. Their responses adequately covered concerns about the novelty of their findings, the role of ATRA in neuronal differentiation, the mechanisms of ferroptosis suppression, and the in vivo relevance of their results. My question regarding n-numbers has been also addressed.

Thank you for the positive evaluation.

- The inclusion of *C. elegans* data supports the in vivo relevance of their findings; further justification of the relevance of this model to human neurodevelopment might strengthen the manuscript. Yet, the authors have addressed potential limitations and outlined future research directions in their discussion.

We are pleased to see that this reviewer appreciates the addition of the *C. elegans* data. We think that the worm data is a piece of the puzzle and that the overall data set shows that ferroptosis can be inhibited by vitamin A, and that ferroptosis must be suppressed to generate proper neurons.

- A final proofread to catch any remaining typographical errors would be helpful. For example, "...neuronal development has to suppress ferroptosis by antioxidants" should be "...neuronal development requires the suppression of ferroptosis by antioxidants." Also, ensure all figure legends are detailed and clear.

We have now proofread the manuscript and feel it is well written.

Overall, the authors have successfully addressed all comments and provided additional data to support their claims. The study offers valuable insights into the role of ferroptosis suppression in neuronal development and the mechanisms of vitamin A in this process.

Thank you for the positive evaluation.

I recommend the revised manuscript for publication after minor typographical corrections and a final proofreading.

Reviewer #2 (Remarks to the Author):

The authors have addressed all my concerns.

Thank you for the positive evaluation.

Reviewer #3 (Remarks to the Author):

This revised manuscript is strengthened and improved related to mode-of-action of retinal, retinol and ATRA, however in vivo relevance (impact) of this mechanism is still not convincingly shown. Experiment in *C. elegans* only shows minor effect, questioning its importance as major anti-ferroptosis mechanism.

We thank the reviewer for the positive evaluation on the mode-of-action of vitamin A. Our study concentrates on the mechanistic details of ferroptosis inhibition during neurodevelopment. The data in *C. elegans* and human brain organoids indeed support our mechanistic study that vitamin A can inhibit ferroptosis. As our institution is committed to the 3R principles (replace, reduce, refine) regarding animal testing, we would like to refrain from in vivo studies that would not add a lot of insight to the mechanistic understanding of this study.